# GENERATIVE PHOTOGRAPHIC CONTROL FOR SCENE-CONSISTENT VIDEO CINEMATIC EDITING

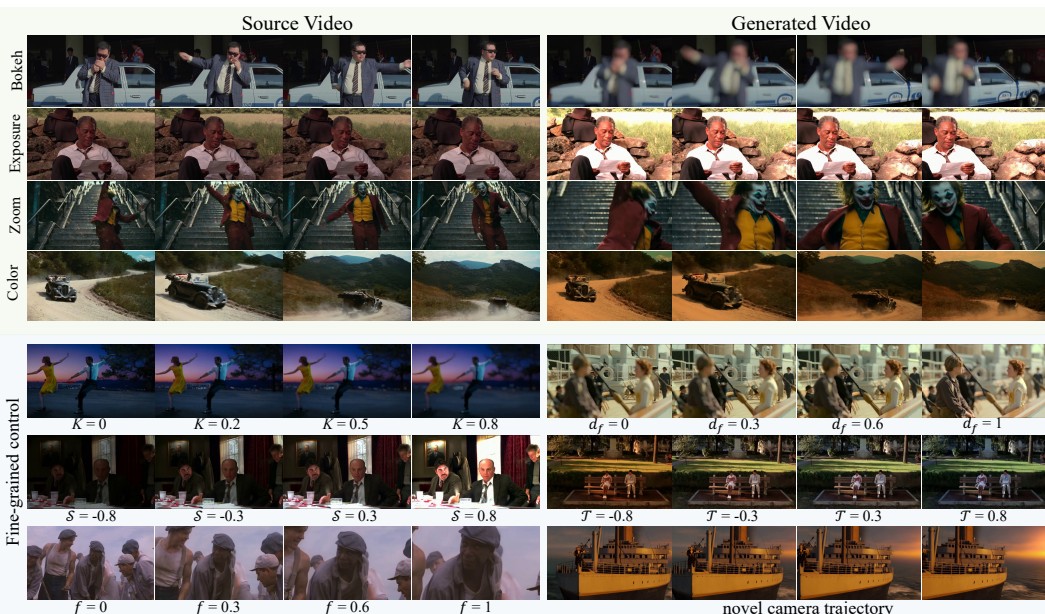

Figure 1: **Examples of fine-grained photographic control with our CineCtrl**. The source video is edited into generated outputs with independently adjusted photographic parameters: bokeh (blur intensity $K$ and refocused disparity $d_f$), exposure (shutter speed $S$), color tone (color temperature $T$), and zoom (focal length $f$), as well as novel camera trajectories. CineCtrl enables precise and disentangled manipulation of these cinematic effects while preserving scene consistency.

## ABSTRACT

Cinematic storytelling is profoundly shaped by the artful manipulation of photographic elements such as depth of field and exposure. These effects are crucial in conveying mood and creating aesthetic appeal. However, controlling these effects in generative video models remains highly challenging, as most existing methods are restricted to camera motion control. In this paper, we propose **CineCtrl**, the first video cinematic editing framework that provides fine control over professional camera parameters (e.g., bokeh, shutter speed). We introduce a decoupled cross-attention mechanism to disentangle camera motion from photographic inputs, allowing fine-grained, independent control without compromising scene consistency. To overcome the shortage of training data, we develop a comprehensive data generation strategy that leverages simulated photographic effects with a dedicated real-world collection pipeline, enabling the construction of a large-scale dataset for robust model training. Extensive experiments demonstrate that our model generates high-fidelity videos with precisely controlled, user-specified photographic camera effects.

## 1 INTRODUCTION

*"A film is never really any good unless the camera is an eye in the head of a poet."*   — Orson Welles

In filmmaking, photographic choices function as a syntax for visual storytelling, shaping how narratives unfold through the camera's lens. For instance, subtle changes in bokeh, color tone, and exposure can recast a neutral frame into one filled with tension, intimacy, or grandeur, guiding viewer attention and conveying emotion. These effects are not merely technical adjustments, but poetic devices, crafted through the skilled manipulation of camera settings, such as aperture, shutter speed, and focal length, that transform the visual narrative into an expressive medium.

Although crucial to cinematic expression, photographic effects remain underexplored in recent generative video models (Blattmann et al., 2023; Chen et al., 2024). While architectural improvements and computational scaling have led to remarkable visual quality, these models generally fall short of offering fine-grained, photographer-level control over key effects. Existing specialized tools (e.g., separate modules for zoom, exposure, or bokeh) can simulate individual effects, but they operate in isolation and often introduce domain gaps or quality degradation when cascaded together. Moreover, such pipelines lack a unified generative prior, making it difficult to preserve scene consistency and cinematic coherence across edits. This limitation directly constrains filmmakers, video editors, and creators who wish to bring professional-level cinematic aesthetics into generative workflows.

Early studies (He et al., 2025; Wang et al., 2024) have explored controllable video generation and editing by employing camera parameters as control signals. However, they mainly focus on controlling camera extrinsics, namely the camera pose and trajectory. For example, ReCamMaster (Bai et al., 2025a) allows users to specify a camera path to re-render an input video from a novel camera trajectory. While viewpoint control offers important capabilities, it leaves untouched the manipulation of explicit professional photographic effects. Although recent work has shown potential in producing visually striking imagery and video (Yuan et al., 2024), its reliance on text-to-visual synthesis limits its suitability for precise and disentangled photographic editing of real-world videos.

In this work, we present **CineCtrl**, the first video cinematic editing framework for explicit and fine control over photographic parameters. Beyond simple trajectory control, CineCtrl enables independent and precise adjustment of cinematic effects such as bokeh, zooming, color temperature, and exposure, as shown in Fig. 1. In general, there are two main obstacles to building our CineCtrl that performs scene-consistent cinematic editing. First, current transformer-based video generators are not inherently designed to separate motion dynamics from photographic stylization. They often entangle spatial-temporal motion cues with appearance-level effects, making independent control of cinematic factors highly challenging. Therefore, straightforward injection methods, e.g., the element-wise addition of the photographic and trajectory control signals with latent tokens, would cause significant visual artifacts with inconsistency. Second, there is a lack of large-scale, well-annotated datasets that capture diverse photographic effects across real-world video domains. Existing datasets either focus on camera trajectories or synthetic renderings, but rarely provide paired variations of professional photographic parameters (e.g., systematic changes in aperture or shutter speed). Without such data, it is difficult to learn disentangled and controllable representations.

To address the architectural limitation, CineCtrl extends a pre-trained text-to-video model (Wan et al., 2025) for novel view synthesis and introduces a dedicated branch for encoding and injecting photographic controls. At its core lies a decoupled cross-attention mechanism that separates camera motion from photographic signals, thereby avoiding control interference and effectively disentangling their respective influences on the generated video, to enable fine-grained, independent control in video generation. In terms of data scarcity, we construct a large-scale dataset combining synthetic and real-world videos. We first design a physically-based simulation method to generate diverse photographic effects, with control signals normalized to a user-friendly range (i.e., [0, 1]). This simulation is applied to the multi-camera synthetic dataset from ReCamMaster (Bai et al., 2025a), forming our primary training set. Then, to enhance real-world generalization, we develop a new data collection pipeline that leverages the simulation method to curate and process 30k cinematic video clips, resulting in a realistic and high-quality dataset. Our training leverages a mixture of both synthetic and real-world data for robust performance.

Our main contributions can be summarized as follows: First, we propose **CineCtrl**, the first generative video cinematic editing model capable of fine photographic effect control through professional

camera parameters. Second, we design a decoupled cross-attention mechanism to effectively disentangle camera motion from photographic signals, ensuring precise and independent control. Third, we introduce a photographic effect simulation method and a real-world dataset pipeline to construct a large-scale dataset for training.

Experimental results demonstrate that our method achieves superior performance against other baselines in producing videos with controllable and fine photographic effects. Ablation studies further confirm the effectiveness of key components in our approach and show the importance of our curated real-world dataset.

## 2  RELATED WORK

**Video Generative Models.**  Video generation has progressed from early framework like GANs (Goodfellow et al., 2014) to the diffusion models (Ho et al., 2020) due to their powerful performance and flexibility in conditioning (Xing et al., 2024; Ho et al., 2022; Singer et al., 2023). Many methods also generate video in a compressed latent space to mitigate the high computational cost (Blattmann et al., 2023; Zhou et al., 2022; He et al., 2022). A breakthrough came with Sora (Brooks et al., 2024), which applies the Diffusion Transformer (DiT) (Peebles & Xie, 2023) architecture to video generation, becoming a dominant paradigm in subsequent video diffusion models (Kong et al., 2024; Wan et al., 2025; Yang et al., 2025). Building on these foundational advances, the generative power of diffusion has fueled a surge in Video-to-Video (V2V) applications (Liu et al., 2024; Ku et al., 2024; Sun et al., 2025). In this paper, we carve out a new direction by proposing a V2V model focused on the explicit control of professional photographic effects, addressing an unexplored area in the literature.

**Camera-Control Video Generation.**  With the development of video generation models, many methods have sought to incorporate more diverse control signals (Guo et al., 2024a; Yin et al., 2023). Camera control has become a particularly active area that aims to generate videos conditioned on specified camera trajectories (Liu et al., 2025; Bai et al., 2025b; Guo et al., 2024b). For example, MotionCtrl (Wang et al., 2024) and CameraCtrl (He et al., 2025) leverage the capabilities of T2V models and training on video-camera pair data to achieve generalized camera control. The pursuit of camera control has also been actively explored in the V2V domain (Van Hoorick et al., 2024; Gu et al., 2025; YU et al., 2025; Wu et al., 2025). Recently, ReCamMaster (Bai et al., 2025a) achieves impressive results through training on a large-scale multi-view dataset. However, the above methods are confined to the control of camera trajectories. Instead, our work extends to a much richer set of photographic parameters, enabling precise control over the photographic effects in the output video.

**Generative Control of Photographic Effects.**  Research on controlling photographic effects in video generation remains significantly underexplored. Some prior works attempt to generate videos with specific storylines via prompt inputs (He et al., 2023; Xiao et al., 2025). However, these methods mainly focus on narrative and semantic consistency, rather than using photographic effects as a storytelling device. While some text-guided models (T2I/T2V) like Camera Setting as Tokens (Fang et al., 2024), Generative Photography (Yuan et al., 2024), and Wan2.2 (Wan et al., 2025) can generate videos with specified photographic camera settings, these methods are inherently unsuitable for editing real-world footage and cannot provide fine control in photographic effects. In contrast, our method introduces the first V2V model with precise photographic control, and proposes a data processing scheme and a real-world data pipeline to build a large-scale training dataset.

## 3  METHOD

Given a source video $V_s$ along with a camera control signal $P$ for photographic effects, a generative video cinematic editing framework is defined as follows:

$$V_t = \mathcal{G}(V_s, P) \tag{1}$$

where $\mathcal{G}$ denotes a generative modeling process under the guidance of selected camera signal $P$ for input video $V_s$. In this work, we instantiate this process within a pre-trained video generative model, utilizing its generative prior to maintain scene consistency and cinematic coherence across edits in target video $V_t$.

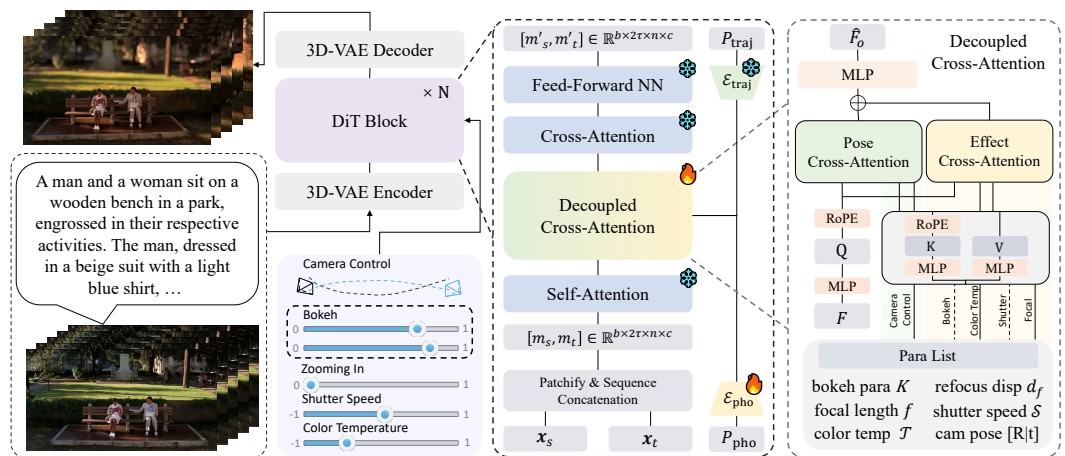

Figure 2: **Overall framework of CineCtrl,** which is built upon the Wan2.1 T2V framework, and extended to a V2V model. To enable camera control, we inject both camera trajectory and photographic parameter signals into the DiT block. Through our proposed *Camera-Decoupled Cross-Attention* mechanism, we disentangle these two signals to achieve accurate and independent control.

## 3.1 CONTROLLABLE VIDEO CINEMATIC EDITING

Our CineCtrl adopts the pre-trained Wan2.1 T2V model (Wan et al., 2025) as the backbone video generator. We introduce both camera extrinsics and professional photographic parameters as control signals to steer input videos. The structural details of CineCtrl are illustrated in Fig. 2.

**Video Encoding.** It begins by encoding a source video $V_s$ and its corresponding target video $V_t$ into latent representations $\boldsymbol{x}_s, \boldsymbol{x}_t \in \mathbb{R}^{b \times \tau \times h \times w \times c}$ by the pre-trained 3D VAE module. Subsequently, the source and noised target latents are patched into $h \times w$ tokens using a patchify operation:

$$\boldsymbol{m}_s = \texttt{Patchify}(\boldsymbol{x}_s), \;\; \boldsymbol{m}_t = \texttt{Patchify}(\hat{\boldsymbol{x}}_t), \;\; \boldsymbol{m}_s, \boldsymbol{m}_t \in \mathbb{R}^{b \times \tau \times n \times c'}, \tag{2}$$

where $n = h \times w$, $\hat{\boldsymbol{x}}_t$ denotes noised target latent, and $c'$ is the channel dimension of the diffusion model. Following ReCamMaster (Bai et al., 2025a), we concatenate $\boldsymbol{m}_s$ and $\boldsymbol{m}_t$ along the sequence dimension to create unified tokens $\boldsymbol{m} = [\boldsymbol{m}_s, \boldsymbol{m}_t] \in \mathbb{R}^{b \times 2\tau \times n \times c'}$ that encapsulate information from both the source and target videos. This sequence is then fed into the DiT backbone to perform the guided diffusion process for cinematic editing.

**Camera Conditioning.** Our cinematography model involves two complementary control signals: $P_{\text{traj}}$ for camera trajectory and $P_{\text{pho}}$ for photographic effects. This dual conditioning enables joint yet disentangled control, allowing trajectory and photographic parameters to be manipulated independently or in combination. Specifically, $P_{\text{traj}} \in \mathbb{R}^{\tau \times 3 \times 4}$ is defined as a sequence of extrinsic matrices that represent the per-frame relative transformation between the camera poses of the source and target videos. If the camera trajectories are set to be identical between input and output videos, $P_{\text{traj}}$ is simply a sequence of identity matrices $[\boldsymbol{I}|\boldsymbol{0}] \in \mathbb{R}^{3 \times 4}$. The parameter $P_{\text{pho}} \in \mathbb{R}^{\tau \times 5}$ provides fine-grained control over the professional photographic effects of the output video. It consists of five elements: bokeh blur parameter $K$, refocused disparity $d_f$, focal length $f$, shutter speed $\mathcal{S}$, and color temperature $\mathcal{T}$, which are used to control the bokeh effect, zooming effect, exposure, and color tone of the output videos. To facilitate user-friendly control, all parameters are normalized to represent relative adjustments from the source video. Parameters $K$, $d_f$, and $f$ are mapped to the range $[0, 1]$, while $\mathcal{S}$ and $\mathcal{T}$ are mapped to $[-1, 1]$. The specific settings for these photographic controls will be detailed in Section 4. To integrate camera control, the $P_{\text{traj}}$ and $P_{\text{pho}}$ signals are respectively passed through two learnable encoders $\mathcal{E}_{\text{traj}}$ and $\mathcal{E}_{\text{pho}}$. These encoders project the parameters into high-dimensional embeddings, matching the channel dimension of the DiT tokens:

$$F_{\text{traj}} = \mathcal{E}_{\text{traj}}(P_{\text{traj}}), \;\; F_{\text{pho}} = \mathcal{E}_{\text{pho}}(P_{\text{pho}}), \;\; F_{\text{traj}}, F_{\text{pho}} \in \mathbb{R}^{\tau \times c'}. \tag{3}$$

Figure 3: **Illustration of the dataset construction.** We generate training pairs by applying our proposed photographic effect simulator to both a synthetic dataset and a high-quality real-world dataset, which we curated from web and movie sources through a shot detection and filtering pipeline.

## 3.2 CAMERA-DECOUPLED CROSS ATTENTION

A straightforward way to inject camera control signals into the DiT backbone would be adding the control features, $F_{\text{traj}}$ and $F_{\text{pho}}$ element-wise to the DiT tokens after dimension expansion. However, our preliminary experiments revealed that this naïve approach induces undesired entanglement between the trajectory and photographic controls, causing visual artifacts in output videos, especially when both camera trajectory and photographic effects are altered simultaneously.

To mitigate cross-signal interference, we propose a novel Camera-Decoupled Cross-Attention layer to inject camera controls, whose architectural details are illustrated in Fig. 2. Specifically, it takes the feature $F$ from the preceding projector layer as input to compute its query $Q$. The key and value pairs are then computed separately from the two control features: $(K_{\text{traj}}, V_{\text{traj}})$ are derived from $F_{\text{traj}}$, and $(K_{\text{pho}}, V_{\text{pho}})$ are derived from $F_{\text{pho}}$. Subsequently, two independent operations (pose and effect cross-attentions) are performed for the trajectory and photographic branches in parallel:

$$O_{\text{traj}} = \text{softmax}\left(\frac{QK_{\text{traj}}^{\top}}{\sqrt{d_k}}\right)V_{\text{traj}}, \quad O_{\text{pho}} = \text{softmax}\left(\frac{QK_{\text{pho}}^{\top}}{\sqrt{d_k}}\right)V_{\text{pho}}. \quad (4)$$

The outputs of the two branches are then summed and passed through a final MLP layer $W_o$:

$$\hat{F}_o = W_o \cdot (O_{\text{traj}} + O_{\text{pho}}). \quad (5)$$

Considering the input camera and photographic signals possess a temporal dimension, we incorporate Rotary Position Embeddings (RoPE) (Su et al., 2024) into the query and key tensors of both branches to preserve temporal information. Furthermore, our Decoupled Cross-Attention layer employs a residual connection, such that:

$$F_o = F + \hat{F}_o. \quad (6)$$

We adopt a zero-initialization strategy for the output projection $W_o$ to stabilize the training process, which allows the model to gradually learn the influence of the control signals. To preserve the original capabilities of the foundational T2V model, we only fine-tune the professional photographic parameter encoder $\mathcal{E}_{\text{pho}}$, the projector layer, and the Camera Decoupled Cross-Attention layer. For the camera trajectory parameter encoder, $\mathcal{E}_{\text{traj}}$, we initialized it with the pre-trained weights from ReCamMaster. The remaining model parameters are kept frozen throughout the training process.

## 4 DATASET

To create necessary training data for video cinematic editing, we design a data curation strategy that comprises two primary components: a method to simulate various fine-grained photographic effects utilizing physical-based models, obtaining source-target video pairs (Section 4.1); and a data acquisition pipeline to curate a high-quality real-world dataset (Section 4.2). A total of 170k synthetic videos and 32k real-world video samples were constructed to form the final training dataset. An overview of the entire data preparation pipeline is provided in Fig. 3. More details are provided in Section A.3-A.4 in the supplementary material.

## 4.1 Photographic Effects Simulation

We generate four types of photographic effects using physically-based simulations (Yuan et al., 2024). All control parameters are normalized to an intuitive range (i.e., [0, 1] and [-1, 1]) for user control.

**Bokeh Effect.** We model the bokeh effect based on the Circle of Confusion (CoC), where the bokeh blur is governed by a magnitude parameter $K$ (related to aperture size) and a refocused disparity $d_f$ (which sets the focal plane). Following existing image-based generative approaches for bokeh (Qin et al., 2025; Wang et al., 2025), our simulation pipeline first estimates disparity maps from an all-in-focus video using Video Depth Anything (Chen et al., 2025) and then applies Bokehme (Peng et al., 2022) to synthesize the effect. For intuitive control, $d_f \in [0, 1]$ and $K \in [0, 1]$, normalized from their physical ranges.

**Zoom Effect.** The zoom effect is controlled by focal length $f$, which inversely determines the Field of View (FoV). We adopt a relative control scheme for the zoom effect to overcome the ill-posed problem of absolute focal length estimation. The effect is simulated by centrally cropping the image according to a target focal length specified within a 24–70mm range. This parameter is then normalized to $[0, 1]$ to serve as the final control signal.

**Exposure.** We simulate exposure changes based on a physical image sensor model (Chi et al., 2023), where the brightness of the image is primarily determined by shutter speed $\mathcal{S}$. To model the non-linear response between shutter speed and image brightness (Debevec & Malik, 1997), we introduce a non-linear multiplier driven by a normalized relative shutter speed $\mathcal{S} \in [-1, 1]$. This provides intuitive, relative control to brighten or darken the output video with respect to the input.

**Color Temperature.** Our color adjustment is based on the black-body radiator model (Fairchild, 2013) where the temperature parameter $\mathcal{T}$ defines the color tone of an image. For intuitive control, we set a base temperature $\text{temp}_s = 6500\text{K}$. A relative parameter $\mathcal{T} \in [-1, 1]$ then shifts the color of the image to warmer or cooler tones by transforming pixel values relative to the base value.

## 4.2 Dataset Construction

To construct the training dataset, we build on the synthetic set from ReCamMaster (Bai et al., 2025a) and apply our photographic effect simulation (Section 4.1) to create video pairs. While this enables control over most effects, we find the bokeh effect, especially the refocused disparity $d_f$, is unreliable due to limited depth diversity in the synthetic data. This domain gap hampers generalization to in-the-wild videos. To overcome this issue, we further curate a large-scale real-world dataset from movies and online videos, chosen for diverse camera motion and depth complexity. Our pipeline has three stages: shot detection, video filtering, and post-processing.

**Shot Detection.** We extract coherent clips without shot boundaries using PySceneDetect (Castellano, 2022), and partition long shots into 81–100 frame segments.

**Video Filtering.** We discard clips that are too short, overly dark, or dominated by facial close-ups (detected with MediaPipe (Lugaresi et al., 2019)), as these hinder reliable bokeh control. We also remove video clips with minimal useful scene information, which are typically characterized by little camera motion and minimal scene dynamics. Such shots offer little valuable information for learning and are removed to enhance the overall quality of the dataset. The detailed evaluation of video information content is provided in Section A.4 in the supplementary.

**Post-processing.** For the remaining clips, we estimate camera poses with MegaSaM (Li et al., 2025), generate captions using Qwen2.5-VL (Bai et al., 2025c), and apply our photographic simulation to form the final paired dataset.

## 5 Experiments

**Implementation Details.** We train our model on the dataset discussed in Section 4, which includes one original set and four with synthetic bokeh, zoom, exposure, and color temperature effects. During training, each sample from the original data is paired with a corresponding video from one of the five subsets, selected randomly at each iteration. For camera trajectory control, we ensure that the

Table 1: Quantitative comparison with other baselines on photographic effect accuracy, video quality, and scene consistency.

| Methods | CorrCoef↑ | | | | LPIPS↓ | CLIP-F↑ | CLIP-V↑ |
|---|---|---|---|---|---|---|---|
| | Bokeh | Zoom | Exposure | Color | | | |
| Text-based baseline (w/o FT) | 0.1959 | -0.1070 | 0.1736 | 0.1440 | **0.4605** | **0.9864** | **0.8910** |
| Text-based baseline (w/ FT) | 0.3204 | 0.1325 | 0.4210 | 0.2470 | 0.6870 | 0.9829 | 0.7431 |
| Stitching baseline | - | - | - | - | 0.7270 | 0.9745 | 0.7297 |
| CineCtrl (ours) | **0.5504** | **0.4550** | **0.5117** | **0.5176** | 0.5360 | 0.9863 | 0.8359 |

Table 2: Quantitative comparison with other baselines on VBench (Huang et al., 2024) metrics.

| Methods | Aesthetic Quality↑ | Imaging Quality↑ | Temporal Flickering↑ | Motion Smoothness↑ | Subject Consistency↑ | Background Consistency↑ |
|---|---|---|---|---|---|---|
| Text-based baseline (w/o FT) | **0.4221** | **0.5068** | 0.9749 | 0.9916 | **0.9253** | 0.9202 |
| Text-based baseline (w/ FT) | 0.3577 | 0.3543 | 0.9811 | 0.9907 | 0.8777 | 0.9204 |
| Stitching baseline | 0.3707 | 0.3431 | 0.9810 | 0.9923 | 0.9006 | 0.9177 |
| CineCtrl (ours) | 0.4017 | 0.4312 | **0.9818** | **0.9925** | 0.9218 | **0.9248** |

camera poses of the video pairs are different in the synthetic dataset. Furthermore, for video pairs all from the original data, we set the photographic control parameters to $K = f = \mathcal{S} = \mathcal{T} = 0$ but randomize $d_f$ within $[0, 1]$. This strategy forces the model to learn that when $K = 0$, the $d_f$ value is irrelevant and should not produce any bokeh effect, thus promoting a robustly disentangled control. The model is fine-tuned for 160k steps on 8 NVIDIA A800 GPUs, using a per-GPU batch size of 1 for a total of 8. We employ a differential learning rate: $1 \times 10^{-4}$ for the camera decoupled cross-attention layer and $1 \times 10^{-5}$ for other modules.

**Evaluation Metrics.** We evaluate our model in terms of photographic effect accuracy, video quality, and scene consistency. For photographic effect accuracy, we calculate the Pearson correlation coefficient (CorrCoef) (Yuan et al., 2024) in each effect. Unlike (Yuan et al., 2024), where the CorrCoef is used to compare the similarity in the trends of change of photographic effects between two videos, we utilize this metric to directly compare the similarity of these effects themselves. please see supplementary materials for details of CorrCoef Metrics. For video quality, we evaluate performance on the widely used VBench (Huang et al., 2024) metrics. Additionally, to evaluate temporal consistency, we compute CLIP-F, which we define as the average CLIP similarity between adjacent frames in the generated video. For scene consistency, we primarily measure the consistency of the scene content between the input and output videos. We employ the frame-wise Learned Perceptual Image Patch Similarity (LPIPS) (Zhang et al., 2018) to calculate the feature-space distance between the output and input videos, as well as CLIP-V (Kuang et al., 2024), which is defined as the CLIP similarity between corresponding frames of the output and input.

**Evaluation Data.** Our primary test set consists of $1,000$ videos randomly sampled from the WebVid (Bain et al., 2021). For the evaluation of camera trajectory control, we follow the evaluation protocol from ReCamMaster (Bai et al., 2025a), testing on 10 different camera trajectories (e.g., pan, tilt, zoom). For photographic effect control, we randomly define two distinct photographic effect parameters for each video. For all defined parameters, $50\%$ of these modify a single effect to test isolated control, while the other $50\%$ apply a complex mixture of effects to test combined control. Notably, our model does not require a depth map as input when rendering the bokeh effect; therefore, we omit depth priors from the test dataset.

## 5.1 COMPARISON

**Baselines.** As there are no existing generative methods that jointly control camera trajectory and professional photographic effects, we devise two types of baselines to validate our approach. The first involves adapting an existing camera trajectory control V2V model to control photographic effects via text prompts. We build upon the open-source, Wan2.1-based pre-trained ReCamMaster model and input the photographic camera parameters as textual prompts (e.g., 'Add no bokeh effect.

Figure 4: **Comparisons with other baselines.** Results demonstrate that CineCtrl achieves fine-grained camera parameter control with high visual quality of output videos.

Table 3: Ablation studies for our proposed key components, including Decoupled Cross Attention (CA), Real Dataset, and Randomize $d_f$.

| Methods | Bokeh CorrCoef↑ | CLIP-V↑ | Imaging Quality↑ | Motion Smoothness↑ | Subject Consistency↑ | Background Consistency↑ |
|---|---|---|---|---|---|---|
| w/o Decoupled CA | 0.4201 | 0.8280 | 0.4129 | 0.9921 | 0.9208 | 0.9226 |
| w/o Real Dataset | 0.5036 | 0.8320 | 0.4268 | 0.9922 | 0.9217 | 0.9238 |
| w/o Randomize $d_f$ | 0.3159 | 0.8013 | 0.3646 | 0.9925 | 0.9120 | 0.9225 |
| Full | **0.5504** | **0.8359** | **0.4312** | **0.9925** | **0.9218** | **0.9248** |

Add zooming effect to input video with a focal length of 0.4'). We evaluate two versions of this type of baseline: (1) a zero-shot version, where we directly use the pre-trained model with the parameter-based text prompts without any fine-tuning, and (2) a fine-tuned version, which is further trained on our proposed dataset using this text-based control. The second type of baseline is not end-to-end, featuring a modular pipeline that simulates our task by stitching together specialized tools. Specifically, we use ReCamMaster and a series of individual, physically-based effect simulation algorithms to apply novel camera trajectories and re-render photographic effects. These algorithms are combined to form a composite baseline for comparison (stitching baseline).

**Quantitative Results.** We evaluate CineCtrl against baselines on photographic control accuracy using CorrCoef for each effect. For each model output under a specific parameter, we generate a pseudo-GT for each effect by applying physical simulation to the source video. We then calculate the CorrCoef between model outputs and these pseudo-GTs to obtain an accuracy score for every single effect. Note that the stitching baseline is excluded since it directly uses these simulations. As shown in Table 1, CineCtrl achieves substantially higher scores than text-driven baselines, confirming the precision of our control. The comparisons of video quality and scene consistency are shown in Table 1 and 2. The stitching baseline degrades quality due to error accumulation across the concatenated components, while our method preserves both fidelity and consistency. Notably, while the text-driven baselines achieve strong results on these metrics, we argue that these measures are misleading in the present context. The low CorrCoef scores and our qualitative results show that the text-based methods fail to render the requested effects correctly. Their outputs remain **mistakenly** similar to the input (e.g., no change in color or exposure), leading to high scores on metrics that favor high input-output similarities.

**Qualitative Results.** Qualitative comparison with the baselines is provided in Fig. 4, visually demonstrating the superiority of our approach. The text-driven baselines struggle to accurately control the photographic effects. Even after being fine-tuned on our dataset, their results remain imprecise. The stitching baseline, which relies on physical simulators to apply these effects, suffers from significant quality degradation due to error accumulation and domain gaps between the cascade

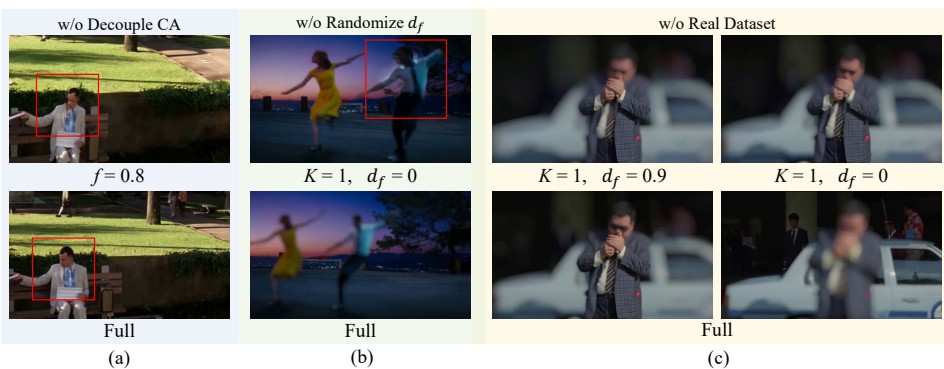

Figure 5: **Qualitative ablation study.** Without Decoupled CA, output videos exhibit noticeable visual artifacts. Besides, control over the bokeh focal plane becomes unreliable when trained without the real-world dataset or using a naïve data parameter setting.

Table 4: User study results indicate that participants prefer our method as of better quality.

| Comparison | Human preference |
|---|---|
| Ours vs. Text-based baseline (w/o FT) | **80.91**% / 19.09% |
| Ours vs. Text-based baseline (w/ FT) | **97.65**% / 2.35% |
| Ours vs. Stitching baseline | **96.15**% / 3.85% |

stages. In contrast, CineCtrl achieves fine-grained control over photographic effects and maintains high visual quality within a single, end-to-end framework. Please refer to our supplementary video for more results.

## 5.2 ABLATION STUDY

We conduct ablation studies on three components: (1) w/o Decouple CA: replacing our Decoupled Cross-Attention with naïve element-wise addition of control features to the DiT tokens. (2) w/o Real Dataset: training without real-world data. (3) w/o Randomize $d_f$: training on a dataset where the parameter $d_f$ for original-to-original pairs were simply set to zero. Quantitative results in Table 3 clearly indicate that the full model, trained with our complete dataset, achieves the best performance. Qualitative results further support these findings. As shown in Fig. 5(a), our decoupling mechanism is crucial for preventing the artifacts that arise from the naïve addition method. Fig. 5(c) visually confirms that the real-world data is essential for the control of the $d_f$ parameter, leading to a more realistic bokeh effect. Finally, Fig. 5(b) shows the visual results for the $d_f$ randomization ablation, demonstrating that our data construction strategy effectively improves $d_f$ control.

## 5.3 USER STUDY

We further conducted a user study to investigate how our method performs in the view of humans when compared with other baselines. We conducted a pairwise comparison using 30 test examples, where participants performed pairwise comparisons between our results and baselines, judging both photographic effect accuracy and video quality. The results of the user study are shown in Table 4, which indicates that our method was preferred by more than 100 participants.

## 6 CONCLUSION

We have presented CineCtrl, the first generative video cinematic editing model for the fine-grained control of professional photographic effects. We proposed a novel Camera-Decoupled Cross-Attention mechanism to inject these control signals, which effectively resolves the issue of interference between camera trajectory and photographic parameters. Furthermore, we also developed a comprehensive data generation strategy for training, combining a physically-based simulation method with a new real-world dataset pipeline. Extensive experiments validate that CineCtrl achieves precise and effective control over the desired photographic effects. For future work, the ca-

pabilities provided by CineCtrl open up possibilities for more intelligent cinematographic systems. A compelling avenue for future research is to build upon our framework by incorporating high-level aesthetic knowledge to automatically determine the optimal camera trajectory and photographic effects for a given scene, paving the way towards automated, cinematic-level video generation.

## ETHICS STATEMENT

All data utilized in this study are obtained exclusively from open-access sources with explicitly defined usage policies. The present work is intended to advance the state of video generative models without introducing ethical or safety risks beyond those already inherent to existing models. Nonetheless, residual concerns, such as dataset biases and the potential for unintended misuse, cannot be entirely ruled out. Accordingly, we underscore the critical importance of rigorous data curation, responsible deployment, and transparent reporting as foundational practices to mitigate these risks and to promote the integrity and reproducibility of research outcomes.

## REPRODUCIBILITY STATEMENT

We place a strong emphasis on reproducibility by offering detailed methodological descriptions designed to support both replication and independent validation. Comprehensive information regarding dataset selection, training strategies, and evaluation protocols is provided to ensure clarity and rigor. Beyond documentation, we are committed to publicly releasing our codebase, pretrained model weights, and accompanying resources. This commitment reflects our broader objective of fostering transparency, promoting open scientific exchange, and empowering the research community to reproduce, scrutinize, and extend our findings in future investigations.

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

# A APPENDIX

## USAGE OF LARGE LANGUAGE MODELS

During the preparation of this manuscript, large language models were employed exclusively as writing assistants. Their role was limited to grammar checking, sentence refinement, and suggesting stylistic alternatives. All substantive content concerning methodology, experimental design, results, and conclusions was conceived and developed entirely by the authors. Outputs generated by the models were critically reviewed, and only human-verified revisions were incorporated into the final version of the text.

## A.1 PRELIMINARY

Our model is built upon the pre-trained Wan2.1 Text-to-Video (T2V) model (Wan et al., 2025), which consists of a 3D Variational Autoencoder (VAE) and a Transformer Diffusion model (DiT). Specifically, for a given input video $V \in \mathbb{R}^{(1+T) \times H \times W \times 3}$, the 3D VAE encoder first encodes the video into a compact latent representation $\boldsymbol{x} \in \mathbb{R}^{\tau \times h \times w \times c}$, where $\tau = (1 + \frac{T}{4})$, $h = \frac{H}{8}$, $w = \frac{W}{8}$. This latent is then noised and processed by the DiT, which consists of $N$ Transformer blocks. For training, we adopt the Flow Matching (FM) framework (Lipman et al., 2023; Liu et al., 2023) as in Wan2.1. This framework learns a velocity field that transports samples from a simple noise distribution to the data distribution via an Ordinary Differential Equation (ODE). The noise latent $\boldsymbol{x}_t$ between a latent data $\boldsymbol{x}_1$ and a random noise $\boldsymbol{x}_0 \sim \mathcal{N}(0, \boldsymbol{I})$ is defined by linear interpolation:

$$\boldsymbol{x}_t = t \cdot \boldsymbol{x}_1 + (1 - t) \cdot \boldsymbol{x}_0. \tag{7}$$

The target velocity field can be written as:

$$\frac{d\boldsymbol{x}_t}{dt} = \boldsymbol{x}_1 - \boldsymbol{x}_0. \tag{8}$$

The diffusion model, denoted as $\boldsymbol{u}_\theta$, is trained to predict this velocity field with the MSE loss function:

$$\mathcal{L}(\theta) = \mathbb{E}_{t, \boldsymbol{x}_0, \boldsymbol{x}_1, c_{\text{txt}}} \| \boldsymbol{u}_\theta(\boldsymbol{x}_t, t, c_{\text{txt}}) - (\boldsymbol{x}_1 - \boldsymbol{x}_0) \|^2, \tag{9}$$

where $c_{\text{txt}}$ represents the conditioning embedding from the text input.

## A.2 DETAILS OF DIT BLOCK

The detailed structure of DiT Block is illustrated in Fig. 2 in the main paper. Within each DiT Transformer block, the input sequence $\boldsymbol{m}$ is first processed by a self-attention layer, followed by an MLP projector. The resulting features are then passed into our Camera-Decoupled Cross-Attention layer (Section 3.2 in the main paper) for camera conditioning, where the control embeddings ($F_{\text{traj}}$ and $F_{\text{pho}}$), representing camera extrinsics and photographic parameters, are injected into the backbone. Subsequently, the camera-conditioned tokens are fed into a standard cross-attention layer. In this layer, text information is incorporated in the form of feature embeddings to enhance the model's semantic understanding of the video content. Finally, the resulting output is processed by the block's concluding Feed-Forward Network (FFN) layer.

## A.3 DETAILS OF PHOTOGRAPHIC EFFECTS SIMULATION

**Bokeh Effect.** The bokeh effect is physically rendered by scattering each pixel with its Circle of Confusion (CoC), where a larger CoC radius yields a more pronounced bokeh blur. The synthesis of the bokeh effect is jointly controlled by two parameters: the bokeh blur parameter $K$ and the refocused disparity $d_f$. The CoC radius $r$ for a given pixel is formulated as:

$$r = K \cdot |d - d_f|, \tag{10}$$

where $d$ represents the disparity of the pixel. As shown in Eq. 10, $K$ is related to the camera aperture size and determines the overall magnitude of the bokeh blur. A larger value of $K$ results in a more intense blur. The parameter $d_f$, on the other hand, controls the position of the focal plane, defining which depth range of the scene appears in sharp focus. When $d_f$ is smaller, regions with smaller

disparity are in focus, effectively moving the focal plane further from the camera. Given an all-in-focus source video, we first generate a dense disparity map for each frame using the Video Depth Anything (Chen et al., 2025). We then synthesize the bokeh effect using Bokehme (Peng et al., 2022), which is based on the principles described above. For a consistent and intuitive control space, all parameters are normalized. The estimated disparity maps are scaled to $[0, 1]$, which naturally constrains $d_f$ to the same range. The parameter $K$, which has a meaningful physical range of $[0, 60]$ ($K = 0$ being no bokeh blur), is also rescaled to a user-friendly $[0, 1]$ range to serve as our final control signal.

**Zoom Effect.** The zoom effect is governed by the focal length $f$, which is inversely related to the Field of View (FoV) for a given image resolution $(h, w)$:

$$\text{FoV} = 2 \cdot \arctan \frac{\sqrt{h^2 + w^2}}{2f} \,. \tag{11}$$

It can be seen that as the focal length $f$ increases, the FOV decreases, producing a zoom-in effect. However, estimating the absolute focal length from a single image is an ill-posed problem. Therefore, we adopt a relative control methodology for the zoom effect. Our method assumes a default source focal length of 24mm (wide-angle) and operates within a 24mm-70mm range. Any specified focal length within this range represents a 'zoom-in' operation relative to the source video. We use the method of cropping the central region of the image to simulate this effect when constructing our video pairs. Specifically, we first calculate the source FOV ($\text{FoV}_s$) using $f_s = 24$mm and the target FOV ($\text{FoV}_t$) for a given target focal length $f_t$. The resolution of the cropped image is determined by the ratio of the target and source FOVs:

$$h' = \left( \frac{\text{FoV}_t}{\text{FoV}_s} \right) \cdot h, \quad w' = \left( \frac{\text{FoV}_t}{\text{FoV}_s} \right) \cdot w \,. \tag{12}$$

The resulting cropped region is subsequently resized back to the original resolution to complete the zoom-in effect simulation. Finally, to serve as the control signal, the focal length parameter $f$, originally specified in the 24-70mm range for dataset generation, is subsequently normalized to the intuitive range of $[0, 1]$.

**Exposure.** The exposure of an image, or its perceived brightness, is fundamentally related to shutter speed $\mathcal{S}$. A higher shutter speed allows more light to enter the camera, resulting in a brighter image. The physical process of image formation from scene light can be expressed by the image sensor model (Chi et al., 2023). This process involves several stages. First, the arrival of photons at each pixel is a random process that follows a Poisson distribution. The incident photons are then converted into electrons by photodiodes, a process with a certain efficiency known as Quantum Efficiency (QE). These electrons are collected in a potential well, which has a finite full well capacity (FWC). If the electron count exceeds this capacity, the charge spills over, causing overexposure. Before the electrical signal is read out, it can be corrupted by noise, including dark current noise $\mu_{\text{dark}}$ (caused by thermal agitation) and read noise $N(0, \sigma_{\text{read}}^2)$ (random electronic perturbations). The collected charge is then converted to a voltage signal, amplified by a conversion gain $\alpha$, and finally digitized to a value between 0-255 by an Analog-to-Digital Converter (ADC). This entire process can be summarized by the following formula:

$$L = \text{ADC} \left\{ \alpha \times \text{Clip} \left\{ \text{Poisson} \left( \mathcal{S} \times \text{QE} \times (H + \mu_{\text{dark}}) \right) \right\} + N(0, \sigma_{\text{read}}^2) \right\} \,, \tag{13}$$

where $H$ is the HDR irradiance of the scene, and Clip denotes the clipping caused by the full well capacity. We can simplify the collected electron count under shutter speed $\mathcal{S}$ as $E(\mathcal{S}) = \text{Poisson} \left( \mathcal{S} \times \text{QE} \times (H + \mu_{\text{dark}}) \right)$.

Like focal length, estimating the absolute shutter speed from a given image is extremely difficult. To enable user control over exposure, we define the shutter speed control signal as a relative change with respect to the input image, constraining its value to the range $[-1, 1]$, where $-1$, $0$, and $1$ correspond to darkening, preserving, and brightening the exposure relative to the source image, respectively. To realize this, we introduce a non-linear exposure multiplier $\mathcal{M}(\mathcal{S})$ as a coefficient to modify the electron count of the input image $E_s$. Based on the observation from (Debevec & Malik, 1997) that exposure is more sensitive in higher irradiance, we formulate the multiplier as an exponential function of the control signal $\mathcal{S}$:

$$\mathcal{M}(\mathcal{S}) = 2^{\epsilon \cdot \mathcal{S}} \,, \tag{14}$$

where $\epsilon$ is a sensitivity hyperparameter. For any given input pixel with color value $c$ and full well capacity $fwc$, we first estimate its original electron count $E_s = (c/255) \cdot fwc$. The target electron count $E_t$ for the output image is then calculated as:

$$E_t = E_s \cdot \mathcal{M}(\mathcal{S}). \tag{15}$$

This target electron count is then processed through the image sensor model (Clip, conversion gain, ADC) to render the final output pixel color. This physical simulation allows us to generate diverse video pairs by sampling the relative shutter speed control $S$ from its normalized $[-1, 1]$ range.

**Color Temperature.** The overall color tone of an image is related to its color temperature, which is physically defined by the Kelvin temperature of a black-body radiator. At low Kelvin values, the emitted light is reddish-yellow, creating a warm tone; at high Kelvin values, the light is bluish, resulting in a cool tone. According to (Fairchild, 2013), the relationship between the RGB values and the Kelvin temperature can be expressed by the following formula:

For temp $\leq 6600$:

$$\mathbf{RGB} = (255, \max(0, 99.47 \cdot \ln(\text{temp}) - 161.12), \max(0, 138.52 \cdot \ln(\text{temp} - 10) - 305.04)), \tag{16}$$

For $6600 < \text{temp} \leq 8800$:

$$\begin{aligned}\mathbf{RGB} = (&0.5 \cdot (255 + 329.7 \cdot (\text{temp} - 60)^{-0.1933}), \\ &0.5 \cdot (288.12 \cdot (\text{temp} - 60)^{-0.1155} + 99.47 \cdot \ln(\text{temp}) - 161.12), \\ &0.5 \cdot (138.52 \cdot \ln(\text{temp} - 10) - 305.04 + 255)),\end{aligned} \tag{17}$$

For temp $> 8800$:

$$\mathbf{RGB} = \left(329.07 \cdot (\text{temp} - 60)^{-0.1933}, 288.12 \cdot (\text{temp} - 60)^{-0.1155}, 255\right). \tag{18}$$

During the dataset construction process, we constrain the Kelvin temperature to the range of $[2000, 10000]$. We also define a normalized relative color temperature parameter $\mathcal{T} \in [-1, 1]$, where $-1$, $0$, and $1$ represent a shift toward a warmer, unchanged, and cooler tone from input videos, respectively. We select a base temperature of $\text{temp}_s = 6500$. The target temperature $\text{temp}_t$ can be calculated from the control signal $\mathcal{T}$:

$$\text{temp}_t = \begin{cases} \text{temp}_s + (\text{temp}_s - 2000) \cdot \mathcal{T}, & -1 \leq \mathcal{T} < 0, \\ \text{temp}_s + (10000 - \text{temp}_s) \cdot \mathcal{T}, & 0 \leq \mathcal{T} \leq 1. \end{cases} \tag{19}$$

We then use Eq. 16-18 to compute the corresponding RGB values for the base and target temperatures, $\text{RGB}_s$ and $\text{RGB}_t$. For an input pixel with color $c$, the corresponding output color $c'$ is calculated as:

$$c' = c \cdot \frac{\text{RGB}_t}{\text{RGB}_s}. \tag{20}$$

The result is subsequently clipped to the valid $[0, 255]$ range. This method is used to construct the color temperature video pairs for our dataset.

## A.4 Details of Real Dataset Pipeline

Our pipeline sources content from movies and online videos, including documentaries, which are specifically chosen for their diverse camera movements and a wide variety of complex depth structures. In this section, we will provide more details about the real-world dataset construction pipeline.

**Video cutting.** The video cutting stage aims to extract temporally coherent short clips from long source videos. The core constraint is the preservation of camera continuity, which requires that no shot boundaries exist within a single training clip. We use PySceneDetect (Castellano, 2022) for robust shot boundary detection, specifically utilizing its detect-adaptive mode for its superior handling of fast cuts. Subsequently, we further partition excessively long sequences into multiple shorter segments to facilitate model training. Following this process, all video clips range from approximately 81 to 100 frames.

**Video filtering.** Video filtering is a pivotal component of our pipeline, designed to distill a high-quality dataset from the vast pool of clips produced by the video cutting stage. This is achieved through a multi-step filtering process based on the following criteria:

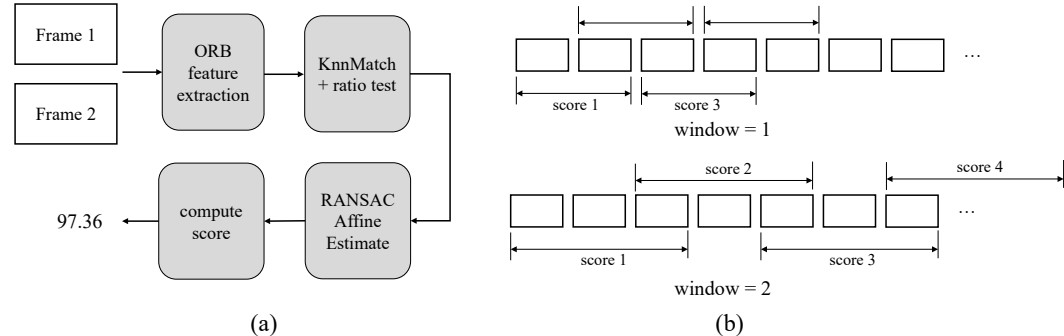

(a)                                                    (b)

Figure 6: **Illustration of the video information filtering.** (a) We compute an information content score by measuring the similarity between two images via a feature matching method. (b) A sliding window approach is then used to calculate the overall information score for the video, where the window size determines the temporal interval between the frames being compared.

*1. Length:* We discard any clips shorter than 81 frames to satisfy the minimum video length requirement for training.

*2. Video Information:* We remove video clips with minimal useful scene information, which are typically characterized by little camera motion and minimal scene dynamics. Such shots offer little valuable information for learning and are removed to enhance the overall quality of the dataset. To quantify the information content of a video clip, we compute the inter-frame changes using a method inspired by video stabilization algorithms (Peng et al., 2024). The overall process is illustrated in Fig. 6(a). For any pair of frames, $I_1$ and $I_2$, we first extract the ORB features (Rublee et al., 2011). A Brute-Force matcher is then used to find the two nearest neighbors in $I_2$ for each feature in $I_1$. We then apply the ratio test, accepting a match only if the ratio of the nearest to the second-nearest neighbor distance is below a certain threshold. After that, we employ RANSAC (Fischler & Bolles, 1981) on these filtered matches to compute the affine transformation between $I_1$ and $I_2$. The magnitude of this transformation is then quantified as a pixel-level displacement score. A larger score signifies a greater change, which we assume correlates with richer information.

To quantitatively score the informational content of a video clip, we calculate the displacement score across different frames of the video. As shown in Fig. 6(b), we partition the video with a window size $w$. For each window, we compute the score between its first and last frames. The average of all these window-based scores is taken as the information score of a video clip. Clips with a score below a predefined threshold are filtered out. However, we empirically found that no single $w$ is optimal: a small $w$ is effective at preserving clips with rich information but may incorrectly discard clips with slow, gradual changes, as the inter-frame difference is minimal. Conversely, a large $w$ can capture slowly evolving scenes but may fail for rapidly changing videos due to the unreliable matches from feature matching. Therefore, we adopt a multi-scale evaluation strategy. We compute two separate scores using both a small and a large window size. A clip is filtered out only if both scores fall below the thresholds. This approach ensures robustness across different motion speeds and significantly minimizes the false rejection of valuable training data.

*3. Facial Close-ups:* Movie videos often contain close-up shots of faces. These shots often feature talking subjects against a heavily blurred background due to the shallow depth of field, which could confound the bokeh control of the model. Specifically, we use the face detection from MediaPipe (Lugaresi et al., 2019) to calculate the size of the facial bounding box in each frame of a video. A clip is then filtered out if the area occupied by the face exceeds a predefined ratio of the total image size.

*4. Luminance:* We filter out overly dark video clips to facilitate the subsequent processes of camera pose estimation and photographic effect synthesis. We convert the video frames to grayscale and compute the average pixel intensity across the entire frame to measure its luminance.

Table 5: Ablation study of Decouple Cross Attention across all photographic effects.

| Methods | CorrCoef↑ | | | |
|---|---|---|---|---|
| | Bokeh | Zoom | Exposure | Color |
| w/o Decoupled CA | 0.4201 | 0.3975 | 0.4920 | 0.5031 |
| CineCtrl (ours) | **0.5504** | **0.4550** | **0.5117** | **0.5176** |

Table 6: Quantitative comparison with separate encoder baseline on photographic effect accuracy, video quality, and scene consistency.

| Methods | CorrCoef↑ | | | | LPIPS↓ | CLIP-F↑ | CLIP-V↑ |
|---|---|---|---|---|---|---|---|
| | Bokeh | Zoom | Exposure | Color | | | |
| Seperate encoder | 0.5152 | 0.4538 | 0.5121 | 0.5088 | 0.5790 | 0.9852 | 0.8429 |
| CineCtrl (ours) | 0.5504 | 0.4550 | 0.5117 | 0.5176 | 0.5360 | 0.9863 | 0.8359 |

## A.5 DISENTANGLEMENT ANALYSIS

We present a comprehensive ablation study evaluating the accuracy metric (CorrCoef) across all photographic effects, as detailed in Table 5. The results demonstrate that removing the Decoupled Cross-Attention mechanism leads to performance degradation across all photographic effects. Given that the test videos contain simultaneous variations in both camera motion and photographic effects, these results demonstrate that our mechanism effectively achieves the disentanglement of control signals.

Theoretically, to achieve a more thorough decoupling effect, we should not limit our approach to distinguishing between camera extrinsics and intrinsics; instead, we could further assign a dedicated encoder for each individual photographic parameter. However, the experimental results in Table 6 and Table 7 demonstrate that processing each photographic parameter separately does not yield a significant improvement in model performance. We think that manipulating camera extrinsics requires the model to synthesize new scene content, aligning this task more closely with **video generation**. Conversely, controlling various photographic effects focuses on modifying the existing input video, aligning more closely with **video editing**. While different intrinsic parameters produce distinct visual effects, they all share the common goal of editing the input image. Consequently, the domain gap between different intrinsics is significantly smaller than the gap between intrinsics and extrinsics. Therefore, even if there is some mutual influence among intrinsics, its impact on the final quality is far less detrimental than the interference caused by extrinsics. On the other hand, leveraging the robust priors of generative models, processing all intrinsic parameters jointly allows the model to learn a holistic representation of photographic style. This fosters an organic unity among different effects, resulting in realistic and consistent outputs even when multiple parameters are adjusted simultaneously.

## A.6 DETAILS OF CORRCOEF METRICS

The CorrCoef used in our experiments is the Pearson Correlation Coefficient, adapted from Generative Photography (Yuan et al., 2024). It measures the linear correlation between the photographic features of model output and a Pseudo-Ground Truth generated via physics-based simulation. For the Bokeh effect, we use the Laplacian operator to quantify the degree of edge blurriness in both the generated video and the physics-simulated reference, and then calculate their correlation. For the zoom effect, we employ feature matching to calculate the relative scaling factor between the generated video and the original video. We then compute the correlation between the scaling factors derived from our model's results and those obtained via physics-based methods. For the exposure effect, we compute the mean grayscale intensity of both the generated video and the pseudo-GT, and then calculate their correlation. For the color temperature, we directly calculate the average RGB

Table 7: Quantitative comparison with separate encoder baseline on VBench (Huang et al., 2024) metrics.

| Methods | Aesthetic Quality↑ | Imaging Quality↑ | Temporal Flickering↑ | Motion Smoothness↑ | Subject Consistency↑ | Background Consistency↑ |
|---|---|---|---|---|---|---|
| Seperate encoder | 0.4023 | 0.4297 | 0.9804 | 0.9928 | 0.9213 | 0.9240 |
| CineCtrl (ours) | 0.4017 | 0.4312 | 0.9818 | 0.9925 | 0.9218 | 0.9248 |

Table 8: Quantitative comparison with ReCamMaster on camera accuracy.

| Methods | RotErr↓ | TransErr↓ |
|---|---|---|
| ReCamMaster | 1.57 | 10.42 |
| CineCtrl (Ours) | 1.62 | 10.14 |

pixel values for both videos to determine their correlation. A CorrCoef closer to 1 indicates that the generated video closely matches the physics engine's result, signifying the model's ability to accurately generate photographic effects. A value near 0 implies no correlation, indicating that the model lacks the capability to generate accurate effects. A value approaching $-1$ represents anti-correlation (e.g., setting a bokeh parameter $K$ near 0 but resulting in excessive blur).

## A.7 CAMERA TRAJECTORY ACCURACY

To quantitatively evaluate the accuracy of camera trajectory control, we use MegaSaM (Li et al., 2025) to estimate the camera pose and calculate the rotation and translation error, denoted as RotErr and TransErr (He et al., 2025; Bai et al., 2025a). We randomly sampled 20 real-world videos for this evaluation, and the comparative results are presented in Table 8. It can be observed that our method maintains a level of trajectory accuracy comparable to the original ReCamMaster model, demonstrating that the integration of photographic control signals does not degrade the performance of the camera motion branch.

## A.8 MORE DETAILS ABOUT USER STUDY

We conducted a user study comprising 30 pairwise comparison trials, featuring a mix of challenging results from both the WebVid test set and our curated real-world videos. The study was hosted on an online website; the screenshot of the interface is shown in Fig. 7. The top-right corner of the interface provides a detailed explanation of each parameter's function to guide the participants' evaluation. The top-left displays the source video, with the corresponding target photographic and camera trajectory parameters shown above. Below are the two videos for comparison: the result from our method and the result from a randomly selected other baseline. To prevent bias, the left-right presentation order of the pair is randomized for each trial. Participants are instructed to select the superior video, with an additional 'Hard to Judge' option for difficult comparisons. The user study is completely anonymous, and it does not involve the collection of any personally identifiable data.

## A.9 MORE RESULTS

In this section, we provide additional visual results. These include comparisons with other baselines (Fig. 8) and a showcase of the effects achieved by our method (Fig. 9). All results demonstrate that our method can achieve high-quality professional photographic control.

Additionally, we experimentally demonstrated the fine-grained control capabilities of our method over photographic effects, with visual results presented in Fig. 10. It can be observed that our method achieves precise control, enabling the effective distinction between effects corresponding to subtle parameter variations, such as the difference between 0.30 and 0.35. However, it is im-

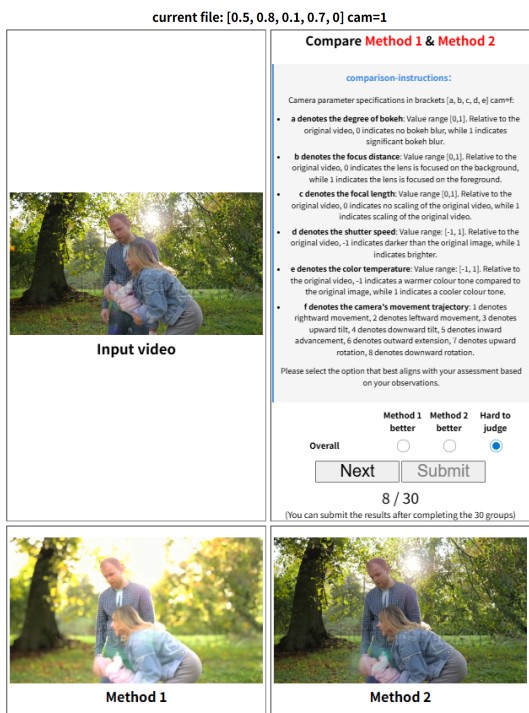

Figure 7: **The website interface for user study.**

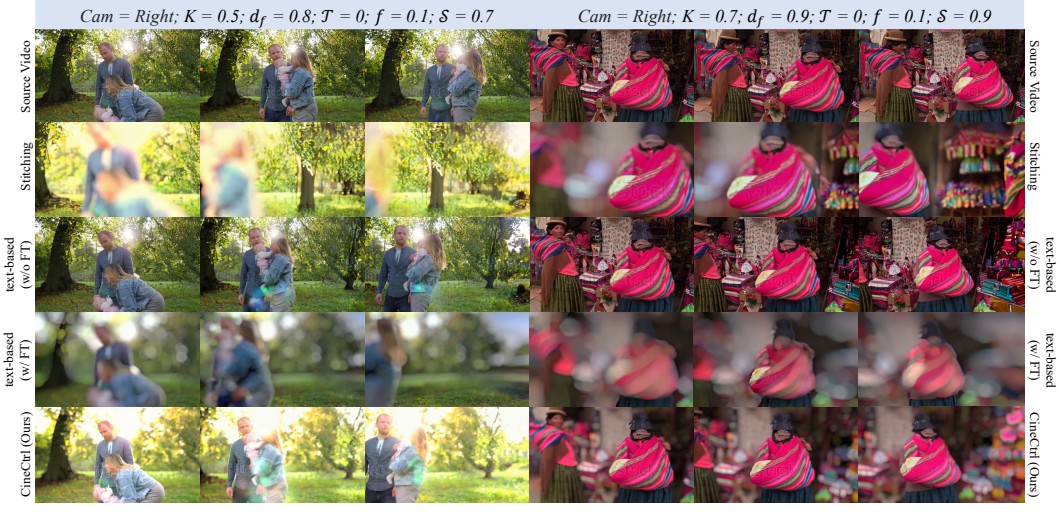

Figure 8: **Comparisons with other baselines.** Results demonstrate that CineCtrl achieves fine-grained camera parameter control with high visual quality of output videos.

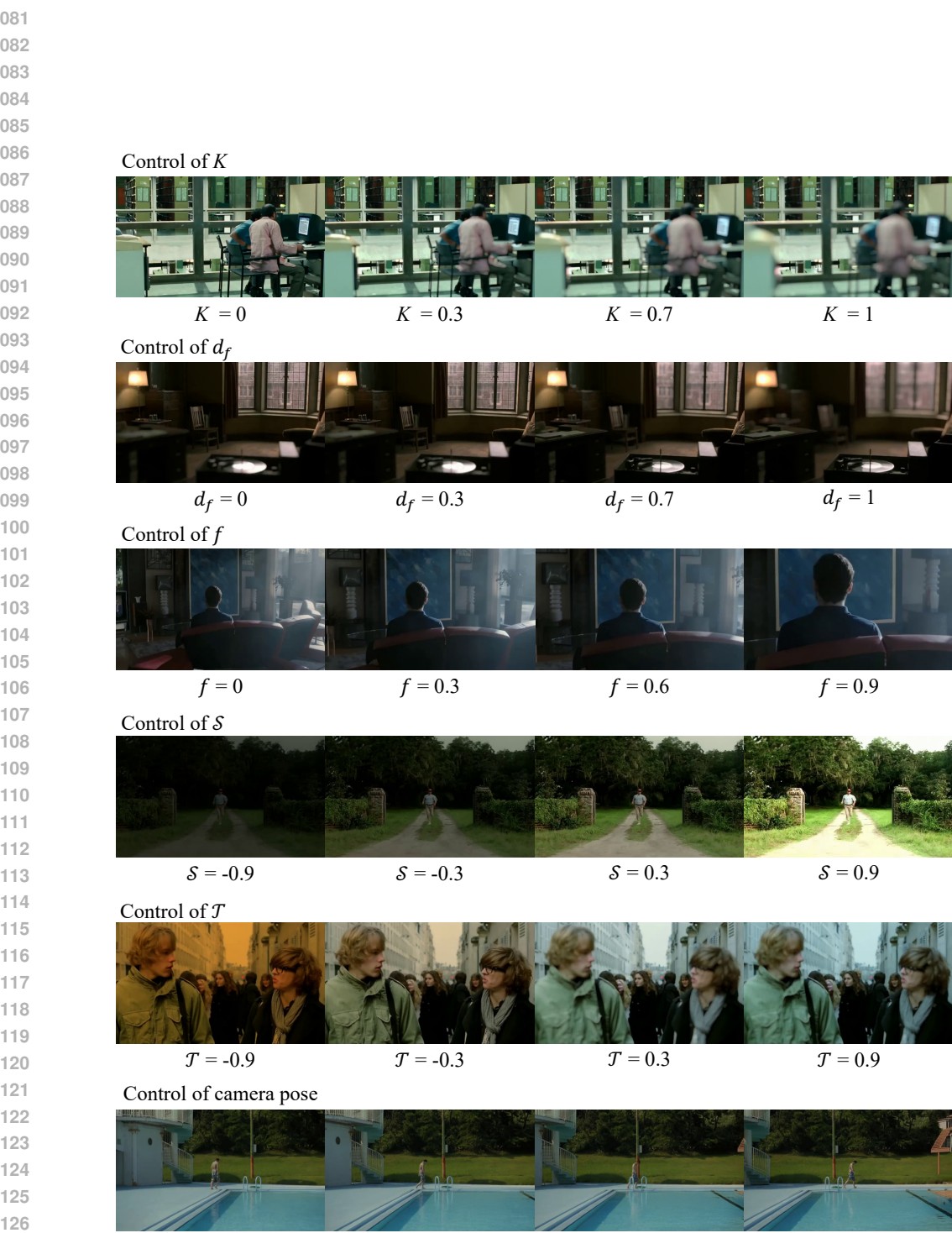

Figure 9: **More visualization of edited videos via CineCtrl.**

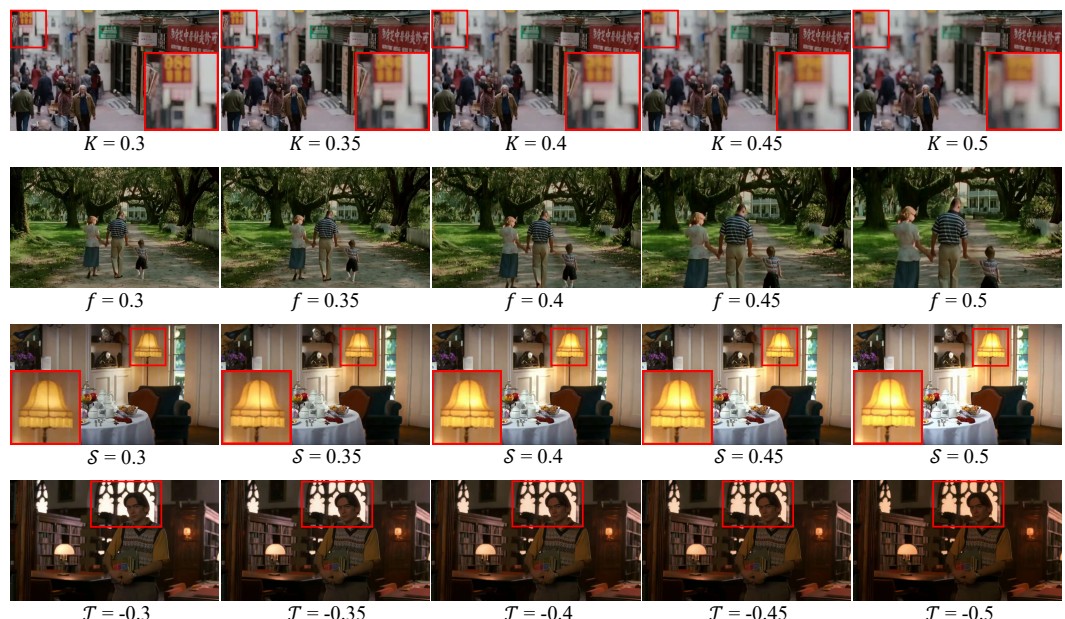

Figure 10: **Visualization of fine-grained control capabilities.**

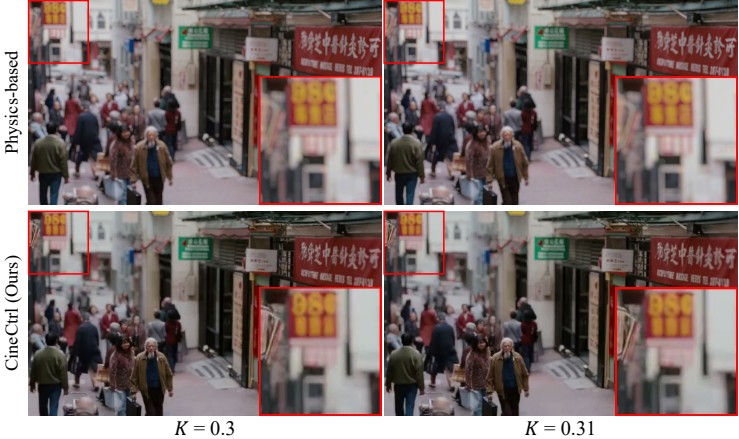

Figure 11: **Fine-grained control limits.**

portant to acknowledge that while our method enables fine-grained control, distinguishing between extremely subtle differences (e.g., 0.30 vs. 0.31) remains challenging. Nevertheless, even when using physics-based rendering methods—which offer the highest level of precision—distinguishing such minute differences is perceptually difficult, as shown in Fig. 11. Therefore, we believe this inherent perceptual limit should not deny the fine-grained control capabilities of our method. Empirically, the high CorrCoef scores in Table 1 demonstrate that our method significantly outperforms text-based baselines in control accuracy, which substantiate our significant advantage in achieving precise and responsive photographic control compared to other methods.

