# OpenReview forum: "Generative Photographic Control for Scene-Consistent Video Cinematic Editing"
_ICLR.cc/2026/Conference — ICLR 2026 Conference Withdrawn Submission_

### Official Review · Reviewer_WY3o · 2025-10-27

**Soundness:** 2
**Presentation:** 3
**Contribution:** 2
**Rating:** 2
**Confidence:** 5

**Summary:**

This paper presents CineCtrl, a video-to-video editing framework that enables fine-grained control over professional photographic parameters (bokeh, exposure, zoom, color temperature) alongside camera trajectories. The authors propose a decoupled cross-attention mechanism to separate camera motion from photographic effects and construct a large-scale training dataset combining synthetic and real-world videos with simulated photographic effects.

**Strengths:**

1. First work to address explicit photographic parameter control in video editing
2. Both synthetic and realistic data construction pipelines are valuable.
3. User study shows clear preference over baselines (Table 4)

**Weaknesses:**

1. Parameters are normalized relative adjustments, requiring users to understand what "K=0.7" means for their specific video. The author should contain these contents in the paper. Besides, the reader does not have a sense of how "good" the metric number represent for, e.g. how much the value of 0.7 is better than 0.5 in the Bokeh metric?
2. This paper claims that it can also control the camera trajectory, but it lacks relative metric? Will the camera trajectory control accuracy worse than the ReCamMaster after finetune?
3. Since all the metric of Table.2 is from VBench, the author should provide corresponding metric values of the base model, e.g. the base video diffusion model and the ReCamMaster.
4. In the qualitative comparisons in Figure 4, there should exist the ground truth results for reference.
5. As stated in Line236, the motivation of Camera-decoupled cross attention is to alleviate the undesired entanglement between the trajectory and photographic controls, thus, in Table 3, only provide the Bokeh CorrCoef metric is not enough.
6. The synthetic data generation process should be more detailed.

**Questions:**

1. Are all the model in the table 1 and 2 finetuned from the same base model? If yes, which model?
2. What is the actual effect strength? The qualitative results show subtle changes - can the model produce dramatic photographic effects?
3. What does the composition of the synthetic data look like? For example, how many samples have a complex mixture of effects and how many samples have isolated control? If a sample has a complex mixture of effects, how does it generated? In the row2 of figure 4, it seems that the stitching will cause the degraded visual quality. If the complex sample is generated using the similar technique, there is an issue that the visual quality of synthetic dataset is low.
4. What does the "Note that the stitching baseline is excluded since it directly uses these simulations." in line416 means?

---

> ### Author Response · Authors · 2025-11-27
> **Response to Reviewer# WY3o (1/2)**
>
> We thank the reviewer for the detailed comments. We address the specific questions below.
>
> > **Concern #1: Normalized Parameters and CorrCoef Metrics**
>
> Normalizing control parameters into the range of $[0, 1]$ or $[-1, 1]$ serves two purposes: it facilitates user-friendly control and addresses the issue that the absolute intrinsic parameters (e.g., exact sensor size or original aperture) of the source video are usually unknown. Our design allows users to control effects based on **perceptual intent** rather than complex physics. Users do not need to understand the specific optical definitions; they simply provide a relative magnitude. For instance, a small $K$ yields a subtle bokeh effect, while a large $K$ produces a strong blur. This applies to all parameters, allowing users to rely on visual perception rather than numerical calculation.
>
> The CorrCoef used in our experiments is the Pearson Correlation Coefficient, adapted from Generative Photography. It measures the linear correlation between the photographic features of our model's output and a Pseudo-Ground Truth generated via physics-based simulation. Taking the Bokeh effect as an example, we use the Laplacian operator to quantify the degree of edge blurriness in both the generated video and the physics-simulated reference, and then calculate their correlation. A CorrCoef closer to $1$ indicates that the generated video closely matches the physics engine's result, signifying the model's ability to accurately generate photographic effects. A value near $0$ implies no correlation, indicating that the model lacks the capability to generate accurate effects. A value approaching $-1$ represents anti-correlation (e.g., setting a parameter $K$ near $0$ but resulting in excessive blur). Therefore, considering an increase from $0.5$ to $0.7$. While $0.5$ indicates the model has learned a basic positive response to the signal, $0.7$ represents a substantial improvement in fidelity and alignment with the target physical effect, confirming that the method follows the user's instructions much more accurately. We will add these clarifications to the revised paper.
>
> > **Concern #2: Camera Trajectory Accuracy**
>
> We appreciate the reviewer's suggestion. To quantitatively evaluate the accuracy of camera trajectory control, we utilized the metrics defined in ReCamMaster. We randomly sampled 20 real-world videos for this evaluation, and the comparative results are presented in the table below. It can be observed that our method maintains a level of trajectory accuracy **comparable** to the original ReCamMaster model, demonstrating that the integration of photographic control signals does not degrade the performance of the camera motion branch.
>
> | Method | RotErr | TransErr |
> | :--- | :---: | :---: |
> | ReCamMaster | 1.57 | 10.42 |
> | Ours | 1.62 | 10.14 |
>
> > **Concern #3: VBench Metric of ReCamMaster**
>
> We clarify that the **'Text-based baseline (w/o FT)'** in Table 2 is generated directly using the pre-trained **ReCamMaster** model (which is built upon the Wan2.1 backbone). Thus, these results serve as the performance reference for the base video generative models requested by the reviewer.
> Comparing our method with this baseline, the results in Table 2 demonstrate that CineCtrl achieves comparable performance on VBench. We acknowledge a minor decrease in 'Aesthetic/Imaging Quality'; however, we argue this is an expected outcome of effective editing. The baseline models often fail to execute the requested photographic effects (as shown in Fig.4 and Fig.8), resulting in outputs that are nearly identical to the high-quality source video, which artificially preserves high quality scores. In contrast, CineCtrl successfully applies these significant visual changes to strictly follow user control while maintaining a high standard of video generation quality.
>
> > **Concern #4: GT in Figure 4**
>
> Fig.4 in the main paper presents the results of our model applied to real-world video clips collected from movies. These results involve simultaneous modifications to both the camera trajectory and photographic effects. Since this task involves synthesizing novel views and effects that do not exist in the original footage, no corresponding Ground Truth (GT) video exists for these specific target parameters in the real world. Therefore, it is inherently impossible to display a pixel-aligned GT reference for these examples. However, the comparative results in Fig.4 and the supplementary video clearly demonstrate that our method significantly outperforms other baselines. Specifically, text-based methods fail to achieve fine-grained control over photographic parameters. The stitching baseline suffers from a noticeable degradation in output video quality. Furthermore, we present the results of a user study comparing different baselines in Table 4, which further confirms the significant advantage of our method.

---

> ### Author Response · Authors · 2025-11-27
> **Response to Reviewer# WY3o (2/2)**
>
> > **Concern #5: Ablation Study**
>
> We appreciate the reviewer's suggestion. To provide a comprehensive evaluation of the disentanglement capability, we extended the ablation study (originally Table 3) to include the CorrCoef metric for all photographic effects under the 'w/o Decoupled CA' setting. The results are presented in the table below. It can be observed that our full model consistently achieves superior performance across every photographic parameter compared to the naive approach ('w/o Decouple CA'). This confirms that the proposed mechanism effectively resolves the interference issues for all control signals.
>
> | Method | Bokeh | Zoom | Exposure | Color |
> | :--- | :---: | :---: | :---: | :---: |
> | w/o Decoupled CA | 0.4201 | 0.3975 | 0.4920 | 0.5031 |
> | **Full** | **0.5504** | **0.4550** | **0.5117** | **0.5176** |
>
> > **Concern #6: Details of Synthetic Data**
>
> We have provided a comprehensive description of the synthetic data generation process in Section A.3 of the Supplementary Material. This section details the physical principles and specific rendering mechanisms for each photographic effect, as well as the parameter normalization strategies employed to ensure user-friendly control.
>
> > **Question #1 Base model**
>
> Both the text-based baselines (in Tables 1 and 2) and our proposed CineCtrl are fine-tuned from the same backbone: the ReCamMaster model. The Stitching Baseline is constructed by stitching the physical rendering algorithms for all photographic effects and ReCamMaster for novel camera views rendering.
>
> > **Question #2 Subtle changes in qualitative results**
>
> We respectfully disagree with the reviewer's observation that the qualitative effects produced by our method are subtle. The visual comparisons in Fig.4 and Fig.8, as well as the Supplementary Video, clearly demonstrate that our method is capable of generating distinct and significant photographic effects, showcasing superior fine-grained control and video quality compared to other baselines. Furthermore, the user study results in Table 4 serve as compelling evidence, confirming that human evaluators perceive a significant advantage in the visual performance and effect accuracy of our method.
>
> > **Question #3 Synthetic data**
>
> Our synthetic dataset is constructed such that each training sample contains exclusively a single photographic effect (e.g., a sample features only bokeh or only exposure adjustment). We do not include samples with complex mixtures of multiple effects in the training set. This isolated construction strategy was intentionally adopted to ensure the high visual fidelity of the training samples, preventing the quality degradation and artifacts that typically arise from stitching multiple simulation steps.
>
> > **Question #4 Meaning of Line 416**
> The statement in Line 416 clarifies the rationale for excluding the stitching baseline from the quantitative evaluation. Calculating the CorrCoef requires a pseudo-Ground Truth (pseudo-GT) for each photographic effect to measure its correlation with the model's generated output. Since real-world videos lack intrinsic ground truth for these edited effects, we employ physics-based rendering methods to generate these pseudo-GTs. However, the stitching baseline itself is constructed using these exact same physics-based simulation algorithms. Consequently, evaluating it using CorrCoef would be tautological—essentially comparing the simulation against itself—which renders the metric meaningless. Therefore, to ensure a valid and fair evaluation, we exclude the stitching baseline from the CorrCoef comparison.

---

> ### Author Response · Authors · 2025-11-28
>
> Dear Reviewer WY3o,
>
> We sincerely thank you for the review and comments. We have posted our response to your initial comments, which we believe has covered your concerns. We are looking forward to your feedback on whether our answers have addressed your concerns or if you have further questions.
>
> Thank you!
>
> Authors

---

### Official Review · Reviewer_qXfZ · 2025-10-30

**Soundness:** 2
**Presentation:** 3
**Contribution:** 3
**Rating:** 6
**Confidence:** 5

**Summary:**

This paper presents CineCtrl, a unified framework for video photography effect editing. It supports editing of bokeh, focal length, exposure, color temperature, and simple camera motion. The method introduces a disentangled attention mechanism to separate the control of camera intrinsics and extrinsics. Both the paper and the supplementary demos show strong visual results.

**Strengths:**

1. This is the first unified framework for video photography effect editing, extending the idea of Generative Photography (Yuan et al. CVPR 2025) to the video domain. It enables joint control over multiple camera parameters — bokeh, focal length, exposure, and color temperature.

2. The proposed approach has strong potential in video editing, generative AI for photography, and visual effects applications.

**Weaknesses:**

1. The disentanglement analysis is not deep enough. In Eq. (5), features from camera intrinsics and extrinsics are directly added — how can this design theoretically achieve disentanglement?

2. There is no clear design for disentangling multiple intrinsics (e.g., focal length vs. exposure). How can these parameters be independently controlled without interference? The paper and supplement lack examples showing the same source video with multiple intrinsics changed simultaneously.

3. The paper does not explain why paired data is needed for training. While this idea was validated in Generative Photography (CVPR 2025), here the motivation is unclear — given that video frames already share scene consistency, the need for paired data should be supported by an ablation.

4. The paper lacks discussion or results on fine-grained control (e.g., bokeh K = 0.3 vs. 0.31), which would demonstrate the precision limit of CineCtrl.

5. The simulation methods used to synthesize the four photographic effects lack necessary citation to Generative Photography (Yuan et al. CVPR 2025).

**Questions:**

Overall, I have a positive impression of this work. The main issues to address are the intrinsic disentanglement design and the fine-grained control demonstration. If the authors can clarify or improve these points, I’d be happy to raise my score.

---

> ### Author Response · Authors · 2025-11-27
> **Response to Reviewer# qXfZ (1/2)**
>
> We appreciate the reviewer's positive assessment and valuable suggestions. We address the specific questions below.
>
> > **Concern #1: Disentanglement Analysis**
>
> We appreciate the reviewer's suggestion. We clarify that signal entanglement typically stems from **non-linear computations** rather than linear operations like addition.
>
> According to the superposition principle, in a linear system, the combined response to multiple signals is equivalent to the sum of individual responses. In our model, the final summation of the extrinsic and intrinsic branches (Eq. 5) is followed by a linear projection. Mathematically, this operation preserves the independence of the signals and does not introduce entanglement. Conversely, the entanglement observed in the naive additive scheme (injecting directly into tokens) arises not merely because the two control signals are added to the backbone tokens, but because the subsequent **non-linear attention mechanism** directly processes these combined features. Since non-linear operations do not satisfy the superposition principle, feeding a mixed signal into a shared attention module inevitably leads to entanglement in the final results. Consequently, the core of our solution is to ensure that different camera signals are processed independently during the non-linear stage (i.e., the attention computation). This motivation led to the design of our Decoupled Cross-Attention mechanism.
>
> > **Concern #2: Intrinsics Disentanglement**
>
> We appreciate the reviewer's valuable feedback. Regarding camera extrinsics: Controlling extrinsics alters the camera trajectory, thereby introducing new scene content in the output video that was not present in the input. This task relies heavily on the model's **generative** capabilities. Regarding camera intrinsics: Modifying photographic parameters (e.g., exposure, bokeh) changes the appearance of existing content without introducing new scene geometry. This relies primarily on the model's **editing** capability. Given the substantial domain gap between *generation* and *editing*, we employ a decoupling strategy to prevent mutual interference between these two distinct processes. While different intrinsic parameters produce distinct visual effects, they all share the common goal of editing the input image. Consequently, the gap between different intrinsics is significantly smaller than the gap between intrinsics and extrinsics. Therefore, even if there is some mutual influence among intrinsics, its impact on the final quality is far less detrimental than the interference caused by extrinsics. On the other hand, leveraging the robust priors of generative models, processing all intrinsic parameters jointly allows the model to learn a **holistic representation** of photographic style. This fosters an organic unity among different effects, resulting in realistic and consistent outputs even when multiple parameters are adjusted simultaneously. The results in Fig. 4 and Fig. 8 of the main paper, as well as the comparative experiments in the supplementary video (where multiple intrinsics are varied simultaneously), demonstrate that our method consistently achieves high-quality, unified results.
>
> We also evaluated a baseline where each intrinsic parameter is encoded separately. The results, shown in the tables below, indicate that using separate encoders for each intrinsic yields no significant improvement over our current approach. Conversely, designing individual encoders for each parameter introduces unnecessary redundancy and complexity to the model. Therefore, we adopted the proposed dual-encoder architecture for injecting camera control signals.
>
> | Method | Bokeh | Zoom | Exposure | Color |
> | :--- | :---: | :---: | :---: | :---: |
> | Separate encoder | 0.5152 | 0.4538 | 0.5121 | 0.5088 |
> | Ours | 0.5504 | 0.4550 | 0.5117 | 0.5176 |
>
> | Method | LPIPS | CLIP-F | CLIP-V |
> | :--- | :---: | :---: | :---: |
> | Separate encoder | 0.5790 | 0.9852 | 0.8429 |
> | Ours | 0.5360 | 0.9863 | 0.8359 |
>
> | Method | Aesthetic Quality | Imaging Quality | Temporal Flickering |
> | :--- | :---: | :---: | :---: |
> | Separate encoder | 0.4023 | 0.4297 | 0.9804 |
> | Ours | 0.4017 | 0.4312 | 0.9818 |
>
> | Method | Motion Smoothness | Subject Consistency | Background Consistency |
> | :--- | :---: | :---: | :---: |
> | Separate encoder | 0.9928 | 0.9213 | 0.9240 |
> | Ours | 0.9925 | 0.9218 | 0.9248 |

---

> ### Author Response · Authors · 2025-11-27
> **Response to Reviewer# qXfZ (2/2)**
>
> > **Concern #3: Need for Paired Data**
>
> We emphasize that paired training data is indispensable for our framework due to the specific requirements of our V2V editing task. Our model functions as a Video-to-Video (V2V) framework, where the primary objective is to inject user-defined photographic effects while **strictly preserving the original scene content** (e.g., object identity, background structure, and motion). Constructing paired data (i.e., a source video and its corresponding effect-rendered target) provides the strong supervision needed to force the model to maintain scene consistency. Without paired data, the model would lack a direct reference for 'invariant content,' leading to severe content drift and hallucinations, making consistent editing impossible.
>
> Besides, our method aims for **precise, fine-grained control** (e.g., mapping a numerical parameter $K=0.7$ to a specific degree of bokeh). Paired data allows the model to learn the exact function mapping between scalar control parameters and their corresponding visual effects. In the absence of paired data, the model would only acquire a coarse perception of various effects, failing to achieve fine-grained control. Therefore, training with paired data is indispensable for our approach.
>
> > **Concern #4: Fine-grained Control**
>
> We appreciate the reviewer's suggestion. In **Fig.10** within the revised manuscript, we demonstrate the fine-grained control capabilities of our method, illustrating its ability to achieve precise control of photographic effects. However, we wish to clarify that while our method supports fine-grained control, visually distinguishing between extremely subtle parameter variations (e.g., the difference between 0.30 and 0.31) remains inherently difficult. It is important to note that such differences are **perceptually indistinguishable** even when using high-precision, physics-based rendering engines, as shown in **Fig.11** in the revised paper. Therefore, we believe that this natural perceptual limit should not negate the fine-grained controllability of our proposed method. The comparative results against text-based baselines in Table 1 of the main paper also demonstrate that our method delivers controllability that is significantly superior to existing baselines.
>
> > **Concern #5: Necessary Citation**
>
> We greatly appreciate the suggestion. We will add the relevant citations to the photographic effects simulation section in the revised version.

---

> ### Author Response · Authors · 2025-11-28
>
> Dear Reviewer qXfZ,
>
> We sincerely thank you for the review and comments. We have posted our response to your initial comments, which we believe has covered your concerns. We are looking forward to your feedback on whether our answers have addressed your concerns or if you have further questions.
>
> Thank you!
>
> Authors

---

### Official Review · Reviewer_GyHW · 2025-10-31

**Soundness:** 2
**Presentation:** 2
**Contribution:** 3
**Rating:** 2
**Confidence:** 3

**Summary:**

This paper proposes a framework for scene-consistent video cinematic editing that enables fine-grained photographic effect control (e.g., exposure, depth of field, color temperature, zoom) on real videos. The method builds upon a pre-trained text-to-video backbone and introduces a camera-decoupled cross-attention module to separate photographic control from camera trajectory control. To train and evaluate the model, the authors construct a hybrid dataset combining simulated and real-world videos with controlled photographic variations. Experiments demonstrate improvements in photographic effect accuracy, video quality, and scene consistency.

**Strengths:**

1. The paper explores an interesting and relatively unexplored direction—adding fine-grained photographic control to video editing.
2. It includes a reasonable data collection pipeline and a simple module design that yields some improvements over baselines.
3. The overall writing and presentation are clear.

**Weaknesses:**

1. Overstated novelty and incomplete related work discussion.
The paper emphasizes its novelty but omits several closely related works, especially in the image domain (e.g., arXiv:2412.02168
), which already demonstrate strong results on similar photographic controls. The paper briefly dismisses these methods as “text-conditioned,” but this difference seems superficial, since both textual and numerical conditions are ultimately embedded as vectors. While I acknowledge the novelty on the video side, I would like to see a clearer explanation of what makes camera-level fine control more challenging in videos compared to images.

2. Limited methodological novelty.
The proposed decoupled cross-attention essentially extends existing camera trajectory conditioning by adding more camera-related dimensions. The design change seems incremental rather than fundamentally new. It would be helpful to clarify what unique insights or training challenges arise from including these additional camera parameters, and how they qualitatively differ from position parameters.

3. Insufficient experimental validation.
a) Baselines. The comparison set is limited. Many related image-based methods could, in principle, be adapted to videos frame by frame, and should be included as baselines rather than only self-implemented ones.
b) Ablations. The paper heavily argues for the effectiveness of the decoupled cross-attention, but lacks comparisons with other conditioning strategies (e.g., text-based injection, additive modulation as in ReCamMaster). The effect on model complexity and parameter count should also be reported.
c) Rationale vs. implementation. While the task itself is meaningful, the implementation seems less justified—since the training data are generated using conventional algorithms, it remains unclear what benefits the learned model provides compared to these traditional methods in terms of efficiency, latency, or effect quality.

**Questions:**

1. Could the authors clarify what makes camera control in video generation fundamentally harder than in image generation, beyond temporal consistency?
2. How does the proposed decoupled cross-attention differ conceptually and functionally from existing trajectory- or motion-conditioned attention mechanisms?
3. Have the authors compared their method against recent image-domain controllable generation works (e.g., [arXiv:2412.02168]) to better contextualize their novelty?
4. Could additional ablations be provided to show the impact of the proposed conditioning design versus simpler baselines?
5. What advantages does the learned control approach provide over traditional camera control methods used in creating datasets in terms of efficiency, latency, or effect quality?

---

> ### Author Response · Authors · 2025-11-27
> **Response to Reviewer# GyHW (1/2)**
>
> We thank the reviewer for the feedback. We address the concerns regarding novelty and baselines below.
>
> > **Concern #1: Differences with Image-based Methods**
>
> We respectfully disagree with the reviewer's assertion that our method lacks distinction from image-level approaches (e.g., Generative Photography). There are **fundamental differences** in task paradigm, precision, and functionality:
>
> 1.  **Task Paradigm (T2I vs. V2V):** Generative Photography is a Text-to-Image (T2I) model. It generates scene images based on text descriptions that combine scene context with photographic effects. Crucially, because T2I generation is stochastic, it cannot guarantee scene identity preservation across multiple generations, nor can it perform **editing** on a specific user-provided input. In contrast, our method is a **Video-to-Video (V2V)** model. Our primary objective is to apply user-specified photographic effects to a *specific* input video while rigorously preserving the original scene content. This requirement for consistency makes our task fundamentally different from that of Generative Photography.
> 2.  **Control Precision:** Text-based control lacks the granularity of our parameter-based control. As demonstrated by the "Text-based baseline" results in **Table 1** (CorrCoef) of our main paper, text-driven control fails to achieve the precise, continuous numerical alignment that our parametric approach delivers.
> 3.  **Functionality:** Our method inherits the **camera trajectory control** capabilities of ReCamMaster. This allows for the joint manipulation of camera movement and photographic effects, a capability inherently absent in image-only frameworks.
>
> Furthermore, the reviewer questioned the difficulties of video methods beyond temporal consistency. We emphasize that temporal consistency itself is the primary barrier distinguishing video methods from image methods; ignoring this factor is inappropriate. Naively extending image methods to video generation results in severe flickering. Furthermore, constructing high-quality video datasets involves unique challenges not found in image datasets, such as handling camera boundaries and abrupt scene cuts.
>
> > **Concern #2: Novelty of Decoupled Cross-Attention**
>
> We disagree with the assessment that our method is merely an incremental extension that 'adds extra parameters'.
>
> Existing methods (e.g., ReCamMaster) typically inject camera control signals by encoding them into tokens and performing **direct element-wise addition** to the backbone features. We observed that naively applying this additive strategy to *both* photographic and trajectory signals simultaneously leads to **undesired entanglement**, resulting in visual artifacts where both trajectory and photographic controls change.
>
> To address this, we fundamentally departed from the ReCamMaster-style injection. We proposed the **Decoupled Cross-Attention** mechanism, which forces the trajectory and photographic signals to be processed through **independent attention paths** with separate Key/Value projections. This ensures the signals remain independent during the non-linear attention calculation, preventing the aforementioned entanglement.
>
> Furthermore, since the control signals possess temporal dimensionality, we incorporated **RoPE** and **adaLN** into the Decoupled Cross-Attention module to enhance responsiveness to temporal information. The effectiveness of this module is empirically validated by the results in **Table 3** and **Fig. 5 (a)** of the main paper.

---

> ### Author Response · Authors · 2025-11-27
> **Response to Reviewer# GyHW (2/2)**
>
> > **Concern #3: Insufficient Experimental Validation**
>
> * **Baselines:** As stated, T2I models like Generative Photography perform a fundamentally different task; they are incapable of editing photographic effects on an existing image/video, making direct comparison infeasible. While there are some existing Image-to-Image (I2I) models specializing in bokeh control, their official code is generally unavailable, preventing direct reproduction. More importantly, these image-based methods inherently lack the capability to synthesize novel views under specified camera trajectories, rendering them fundamentally incomparable to our framework, which integrates explicit camera motion control. Furthermore, it is a well-established empirical observation that applying such image-based methods in a frame-by-frame manner fails to preserve temporal consistency, inevitably leading to flickering artifacts in the output video.
> * **Ablations:** Comparisons with alternative conditioning strategies are **already included** in the paper:
>     * **Text-based baseline (Tables 1 & 2):** Represents controlling effects via textual prompts. Our method demonstrates superior fine-grained control.
>     * **w/o Decouple CA (Table 3):** Represents the strategy of additive modulation (as in ReCamMaster). Our method effectively resolves the entanglement issue present in this baseline.
> * **Traditional Methods:** As discussed in Lines 67-72 of the paper, while traditional algorithms can model individual effects with high quality, the domain gap between different effects leads to significant quality degradation and scene inconsistency when they are cascaded and stitched together. We also validated this claim in Tables 1 & 2 using the **'Stitching Baseline'**, which is exactly the cascade of these traditional physical simulation algorithms. The results demonstrate that CineCtrl achieves superior video quality and scene consistency compared to this traditional pipeline, proving that our end-to-end learned model successfully harmonizes these effects rather than merely replicating the flaws of the simulation tools.
>
> > **Questions:**
>
> The points raised in the 'Questions' section have been discussed in our detailed responses above. We will explicitly highlight these discussions in the revised manuscript to ensure clarity.

---

> ### Author Response · Authors · 2025-11-28
>
> Dear Reviewer GyHW,
>
> We sincerely thank you for the review and comments. We have posted our response to your initial comments, which we believe has covered your concerns. We are looking forward to your feedback on whether our answers have addressed your concerns or if you have further questions.
>
> Thank you!
>
> Authors

---

### Official Review · Reviewer_4wEh · 2025-11-01

**Soundness:** 3
**Presentation:** 3
**Contribution:** 2
**Rating:** 4
**Confidence:** 4

**Summary:**

The paper proposes CineCtrl, a video-to-video (V2V) cinematic editing framework that enables fine-grained, scene-consistent control over professional photographic parameters—including bokeh, exposure, color temperature,  focal length, and also camera trajectory.

The core contributions are as follows:

1. Beyond camera extrinsics, the paper introduces a more comprehensive set of photographic control parameters for video translation, covering professional attributes such as bokeh, exposure, color temperature, and focal length.

2. It develops a data preparation pipeline that constructs both synthetic and real paired datasets of “the same scene under different photographic settings,” enabling supervised learning of controllable cinematic effects.

**Strengths:**

### Originality
1. The paper explores an under-addressed aspect of controllable video translation—photographic controls such as bokeh, exposure, color temperature, and focal length—rather than only camera trajectories. This significantly broadens the scope of controllable video translation.
2. The data side is also non-trivial: physically inspired simulation for 4 kinds of photographic effects + a real-data curation pipeline so the model doesn’t overfit to synthetic depth or simple zooms.

### Technical Quality
1. The method is built on top of a strong, modern video diffusion backbone (DiT-based) and reuses a proven camera-control encoder (from ReCamMaster) for the trajectory branch, which makes the engineering story credible.
2. Evaluation considers both: (i) effect accuracy (CorrCoef for bokeh/zoom/exposure/color) and (ii) scene/video consistency (LPIPS, CLIP-V, VBench metrics), which is appropriate.

### Clarity
1. The paper has a clear structure and is easy to read.
2. Figure 2 very clearly shows where the two control streams enter the DiT block.

### Significance
1. This work is valuable, as it enables a single model to modify real videos with precise photographic adjustments—for example, producing the same scene with a tighter focal length, warmer tone, shallower depth of field, or a shifted camera trajectory.

**Weaknesses:**

1. **Limited Novelty and Missing Comparison.**
   The proposed *Camera-Decoupled Cross-Attention* is conceptually similar to existing decoupled cross-attention mechanisms such as IP-Adapter (Sec. 3.2.2), but the paper does not clearly explain the differences or cite related works, reducing the perceived originality.

2. **Data Pipeline Reliability.**
   The data synthesis pipeline depends heavily on depth estimation (“Video Depth Anything”) and bokeh simulation, both of which are error-prone and can fail on thin structures or dynamic scenes, potentially limiting data quality and model robustness.

3. **Lack of Quantitative Disentanglement Analysis.**
   Although the decoupled attention empirically outperforms naïve fusion, the paper does not provide quantitative evidence of control independence (e.g., verifying that changing exposure does not affect motion or depth-of-field), leaving disentanglement only qualitatively demonstrated.

**Questions:**

1. **Originality Concern.**
   My main concern lies in the originality of this work. Although extending controllable video generation to include richer photographic parameters is meaningful, the core method — *Camera-Decoupled Cross-Attention* — appears highly similar to prior approaches. The authors should clearly articulate how their design differs from existing methods and what novel insights it provides.

2. **Control Resolution.**
   The paper states that all controllable parameters are normalized to ([0,1]) or ([-1,1]). How fine is the actual control in practice? For instance, for the bokeh parameter, can users reliably perceive a difference between 0.3 and 0.35, or is the control effectively quantized into only a few visually distinct levels?

3. **Design of Dual Encoders.**
   The model employs two learnable encoders — one for camera extrinsics and another for other photographic parameters. What is the motivation for this two-encoder design? Why not assign a separate encoder to each controllable parameter for potentially finer disentanglement and flexibility?

---

> ### Author Response · Authors · 2025-11-27
> **Response to Reviewer# 4wEh (1/3)**
>
> **Response to Reviewer 4wEh**
>
> We sincerely thank the reviewer for the constructive feedback and for recognizing the value of our work on photographic control and the data pipeline. We address your concerns and questions below.
>
> > **Concern #1: Novelty of Decoupled Cross-Attention**
>
> We thank the reviewer for pointing out the structural resemblance. While our **Camera-Decoupled Cross-Attention** shares a high-level parallel structure with IP-Adapter to inject signals, the two methods differ fundamentally in motivation, optimization strategy, and temporal architectural design. We clarify these distinctions as follows:
>
> * **Motivation and Optimization Paradigm:**
>     * **IP-Adapter (Injection):** Its primary goal is to inject a new modality (image features) into a pre-existing text-to-image model. Crucially, it adopts a decoupled training strategy where the original text branch is **frozen** to preserve the pretrained priors, training only the new image branch. The two signals are treated hierarchically (base vs. adapter).
>     * **CineCtrl (Disentanglement):** Our goal is to resolve the **physical coupling** between two conflicting video control signals: Camera Extrinsics (geometric motion) and Photographic Intrinsics (optical effects). We observed that naively adding these features together leads to "signal fighting," causing visual artifacts in output videos (as shown in Fig. 5(a), where the zooming effect and camera motion change simultaneously). Therefore, our mechanism is designed for **disentanglement**, not just injection. Unlike IP-Adapter, we employ **joint optimization**: both attention branches are trained simultaneously as equal peers. We do not freeze one to accommodate the other; instead, the model learns to balance and separate the camera motion and optical effect flows dynamically.
>
> * **Architectural Design:**
>     * Unlike the standard cross-attention in IP-Adapter (which typically handles static images and prompts), our control signals (trajectory and photographic parameters) are **time-variant sequences**. To strictly align these controls with the video temporal tokens, we integrate **Rotary Positional Embeddings (RoPE)** into the Query and Key of our attention mechanism. Also, following the architecture of Wan2.1, we incorporate **Adaptive Layer Normalization (adaLN)** to enhance the model's responsiveness to time-series inputs. Furthermore, we employ a **Zero-Initialized MLP ($W_o$)** with a residual connection. This ensures that the training starts from a neutral state, allowing the control signals to be learned progressively without destabilizing the backbone learning, which is critical for maintaining video coherence and quality.
>
> > **Concern #2: Data Pipeline Reliability**
>
> We appreciate the reviewer raising this point. We agree that tools like *BokehMe* and *Video Depth Anything* have limitations in complex scenes. However, we would like to clarify that the primary objective of our work is not to achieve pixel-level precision in bokeh rendering for every corner case. Instead, our goal is to develop a unified generative framework capable of synthesizing a wide range of photographic effects, focusing on the **perceptual realism** of the overall bokeh effect.
>
> We adopted the 'Depth Estimation + BokehMe' pipeline primarily because it successfully yields realistic results in the majority of scenarios. Furthermore, there is a notable scarcity of high-quality, real-world bokeh datasets in the community, and synthetic bokeh generated by rendering engines often lacks photorealism. Consequently, most image-based generative approaches for bokeh, such as DiffCamera [1] and CamEdit [2], employ this exact pipeline to construct their training datasets. While we acknowledge the potential artifacts in complex scenes, this pipeline currently represents the optimal trade-off for large-scale data generation.
>
> Moreover, qualitative results in the main paper and the supplementary video demonstrate that our method produces high-quality, realistic bokeh effects that outperform baseline methods (which rely on simply stitching traditional algorithms like BokehMe). This evidence validates that, despite being trained on datasets generated by traditional algorithms, our unified framework successfully and organically integrates multiple photographic effects, ensuring robustness even when complex effects are combined.
>
> **References:**
>
> [1] DiffCamera: Arbitrary Refocusing on Images, SIGGRAPH Asia 2025
>
> [2] CamEdit: Continuous Camera Parameter Control for Photorealistic Image Editing, NeurIPS 2025

---

> ### Author Response · Authors · 2025-11-27
> **Response to Reviewer# 4wEh (2/3)**
>
> > **Concern #3: Disentanglement Analysis**
>
> We present a comprehensive ablation study evaluating the accuracy metric (**CorrCoef**) across all photographic effects, as detailed in the table below. The results demonstrate that removing the Decoupled Cross-Attention mechanism leads to performance degradation across all photographic effects. Given that the test videos contain simultaneous variations in both camera motion and photographic effects, these results demonstrate that our mechanism effectively achieves the disentanglement of control signals.
>
> | Method | Bokeh | Zoom | Exposure | Color |
> | :--- | :---: | :---: | :---: | :---: |
> | w/o Decoupled CA | 0.4201 | 0.3975 | 0.4920 | 0.5031 |
> | **Full** | **0.5504** | **0.4550** | **0.5117** | **0.5176** |
>
> Furthermore, in response to the reviewer's suggestion, we performed a quantitative disentanglement analysis by varying specific parameters (either camera motion or one of the photographic effects) while measuring the performance metrics of the others. Specifically, we randomly sampled 50 test videos from the WebVid dataset and conducted two sets of experiments:
> 1.  **Experiment 1:** We modified the camera trajectory and measured the variations in metrics for the photographic effects.
> 2.  **Experiment 2:** We varied the exposure effect and evaluated the metrics for other photographic effects and camera motion.
>
> The quantitative results are presented in the tables below. As observed, altering the camera trajectory does not significantly impact the generation of photographic effects. Similarly, modifying the exposure effect yields no significant influence on the performance of other effects or camera motion control. These data provide quantitative evidence validating the decoupling capability of our method.
>
> *Table: Effect of Camera Change on Photographic Metrics (CorrCoef)*
> | Camera Change | Bokeh | Zoom | Exposure | Color |
> | :--- | :---: | :---: | :---: | :---: |
> | Pan Right | 0.4751 | 0.4329 | 0.4476 | 0.4789 |
> | Tilt Up | 0.4732 | 0.4330 | 0.4396 | 0.4750 |
> | Arc Left | 0.4705 | 0.4482 | 0.4336 | 0.4841 |
>
> *Table: Effect of Exposure Change on Other Metrics*
> | Exposure Change | Bokeh (CorrCoef) | Zoom (CorrCoef) | Color (CorrCoef) | RotErr (Cam Acc) | TransErr (Cam Acc) |
> | :--- | :---: | :---: | :---: | :---: | :---: |
> | -0.5 | 0.5201 | 0.4476 | 0.4831 | 1.82 | 9.73 |
> | 0 | 0.5317 | 0.4463 | 0.4954 | 1.84 | 9.85 |
> | 0.5 | 0.5243 | 0.4428 | 0.4897 | 1.88 | 9.76 |
>
> > **Question #1: Control Resolution**
>
> Our framework normalizes all control parameters into continuous spaces (i.e., $[0, 1]$ or $[-1, 1]$), theoretically allowing users to input any arbitrary numerical value within this space. In the revised paper, we demonstrate the granularity of our model's control capabilities. As illustrated in **Fig. 10**, our model achieves fine-grained control over photographic effects, rather than simply quantizing them into a few discrete levels.
>
> However, it is important to acknowledge that distinguishing between extremely subtle differences (e.g., 0.30 vs. 0.31) remains challenging. Nevertheless, even when using physics-based rendering methods—which offer the highest level of precision—distinguishing such minute differences is **perceptually difficult** (as shown in **Fig. 11** in the revised paper). Therefore, we believe this inherent perceptual limit should not negate the fine-grained control capabilities of our method. Empirically, the high CorrCoef scores in Table 1 demonstrate that our method significantly outperforms text-based baselines in control accuracy.

---

> ### Author Response · Authors · 2025-11-27
> **Response to Reviewer# 4wEh (3/3)**
>
> > **Question #2: Design of Dual Encoders**
>
> * **Functional Distinction:** Manipulating camera extrinsics requires the model to synthesize new scene content, aligning this task more closely with **scene generation**. Conversely, controlling various photographic effects focuses on modifying the existing input video, aligning more closely with **video editing**. Therefore, we employ two separate encoders to encode these signals respectively. This design facilitates decoupled control, allowing the model to handle the distinct requirements of extrinsic generation and photographic parameter editing effectively.
> * **Efficiency:** Regarding the photographic effect encoder, its essential function is to project low-dimensional photographic parameters into a high-dimensional space to align with the dimensions of the backbone network's tokens. Consequently, employing a separate, dedicated encoder for each individual photographic effect yields minimal performance gains.
>
> Following the reviewer's suggestion, we conducted an experiment where a separate encoder was assigned to each photographic effect. The results (tables below) indicate that this configuration does not lead to significant performance improvements. However, such a design introduces unnecessary redundancy and complexity to the model. Therefore, we use our dual-encoder architecture.
>
> | Method | Bokeh | Zoom | Exposure | Color |
> | :--- | :---: | :---: | :---: | :---: |
> | Separate encoder | 0.5152 | 0.4538 | 0.5121 | 0.5088 |
> | Ours | 0.5504 | 0.4550 | 0.5117 | 0.5176 |
>
> | Method | LPIPS | CLIP-F | CLIP-V |
> | :--- | :---: | :---: | :---: |
> | Separate encoder | 0.5790 | 0.9852 | 0.8429 |
> | Ours | 0.5360 | 0.9863 | 0.8359 |
>
> | Method | Aesthetic Quality | Imaging Quality | Temporal Flickering |
> | :--- | :---: | :---: | :---: |
> | Separate encoder | 0.4023 | 0.4297 | 0.9804 |
> | Ours | 0.4017 | 0.4312 | 0.9818 |
>
> | Method | Motion Smoothness | Subject Consistency | Background Consistency |
> | :--- | :---: | :---: | :---: |
> | Separate encoder | 0.9928 | 0.9213 | 0.9240 |
> | Ours | 0.9925 | 0.9218 | 0.9248 |

---

> ### Author Response · Authors · 2025-11-28
>
> Dear Reviewer 4wEh,
>
> We sincerely thank you for the review and comments. We have posted our response to your initial comments, which we believe has covered your concerns. We are looking forward to your feedback on whether our answers have addressed your concerns or if you have further questions.
>
> Thank you!
>
> Authors

---

### Note · Authors · 2026-02-27

I have read and agree with the venue's withdrawal policy on behalf of myself and my co-authors.

---

### Meta-Review · Area_Chair_6mrE · 2026-01-07

**Summary:**

The AC carefully read the paper and the full discussion. The submission received mixed reviews (initial scores: 4, 2, 6, 2). Reviewers generally agreed that the paper targets an interesting and relatively underexplored problem—bringing fine-grained photographic control to video editing—and appreciated the proposed data pipeline. However, the main concerns center on limited novelty and insufficient experimental validation, which raise questions about robustness, generality, and practical usability. As a result, it is still unclear whether the proposed method and dataset pipeline deliver clear, meaningful benefits for current or future cinematic video editing models. With the overall scores trending toward rejection and the core issues seeming unlikely to be resolved through discussion, I am inclined to recommend rejection.

**Reviewer Concerns:**

Some concerns regarding intrinsic disentanglement and the need for paired training data (reviewer qXfZ), as well as the explanation of the method and details of the synthetic data pipeline (WY3o), have been addressed. However, several non-negligible issues remain:

Two reviewers (e.g., 4wEh and GyHW) noted the limited novelty. In particular, the proposed decoupled cross-attention appears to be an incremental extension of existing camera-trajectory conditioning, primarily by introducing additional camera-related conditioning dimensions, rather than a fundamentally new design. In addition, the manuscript omits several closely related works.

There are also concerns about insufficient experimental validation. Many related image-based methods could, in principle, be adapted to video in a frame-by-frame manner and should be included as baselines, rather than relying mainly on self-implemented comparisons. The authors also claim that no public implementations are available, but this appears inaccurate, and comparisons to professional AI editing tools (e.g., Runway) could further strengthen the evaluation.

Overall, it remains unclear whether the proposed approach delivers clear or meaningful benefits for current or future cinematic video models.

**Reviewer Scores:**

Reviewer 4wEh might at most raise the score slightly, but would likely still lean toward rejection, primarily due to novelty concerns.

Reviewer GyHW is expected to keep the current score, citing limited novelty.

Reviewer qXfZ will likely increase the score, as the issues around intrinsic disentanglement design and the fine-grained control demonstrations have been addressed.

Reviewer wy3o is also likely to maintain the original score.

---

### Decision · Program_Chairs · 2026-01-26

Reject